# A Comparative Overview of Technological Advances in Fall Detection Systems for Elderly People

**DOI:** 10.3390/s25247423

**Published:** 2025-12-05

**Authors:** Omar Flor-Unda, Rafael Arcos-Reina, Cristina Estrella-Caicedo, Carlos Toapanta, Freddy Villao, Héctor Palacios-Cabrera, Susana Nunez-Nagy, Bernardo Alarcos

**Affiliations:** 1Ingeniería Industrial, Facultad de Ingeniería y Ciencias Aplicadas, Universidad de las Américas, Quito 170125, Ecuador; 2Escuela de Fisioterapia, Facultad de Ciencias de la Salud, Universidad de Las Américas, Quito 170125, Ecuador; rafael.arcos@udla.edu.ec; 3Ingeniería en Telecomunicaciones, Departamento de Eléctrica, Electrónica y Telecomunicaciones, Universidad de las Fuerzas Armadas, Sangolquí 171103, Ecuador; estrellacristina03@hotmail.com; 4Facultad de Ciencias Sociales y la Salud, Universidad Estatal Península de Santa Elena, La Libertad 240204, Ecuador; ctoapanta@upse.edu.ec (C.T.); fvillao@upse.edu.ec (F.V.); 5Facultad de Ciencias de la Salud, Universidad Espíritu Santo, Samborondón 092301, Ecuador; hpalacioscabrera@gmail.com; 6Andrews University, Berrien Springs, MI 49104, USA; 7Physiotherapy Unit, Department of Nursing and Physiotherapy, Faculty of Medicine and Health Sciences, University of Alcalá, 28805 Alcalá de Henares, Spain; susana.nunez@uah.es; 8Health Technology Integration Research Group (GITES), The Castilla-La Mancha Health Research Institute, 45071 Toledo, Spain; 9Polytechnic School, University of Alcalá, 28805 Alcalá de Henares, Spain; bernardo.alarcos@uah.es

**Keywords:** fall detection, elderly, wearable sensors, artificial intelligence, machine learning, inertial measurement units

## Abstract

**Highlights:**

**What are the main findings?**

AI, IoT, and wearable devices improve accuracy in fall detection.Technology enables smarter, more preventative systems.

**What is the implication of the main finding?**

Intelligent systems reduce fall response times.Predictive technologies lower unnecessary hospitalisations.

**Abstract:**

Population ageing is a growing global trend. It was estimated that by 2050, people over 60 years of age will represent 35% of the population in industrialised countries. This context demands strategies that incorporate technologies, such as fall detection systems, to facilitate remote monitoring and the automatic activation of risk alarms, thus improving quality of life. This article presents a scoping review of the leading technological solutions developed over the last decade for detecting falls in older adults, describing their principles of operation, effectiveness, advantages, limitations, and future trends in their development. The review was conducted under the PRISMA^®^ methodology, including articles indexed in SCOPUS, ScienceDirect, Web of Science, PubMed, IEEE Xplore and Taylor & Francis. There is a predominance in the use of inertial systems that use accelerometers and gyroscopes, valued for their low cost and wide availability. However, those approaches that combine image analysis with artificial intelligence and machine learning algorithms show superiority in terms of accuracy and robustness. Similarly, progress has been made in the development of multisensory solutions based on IoT technologies, capable of integrating information from various sources, which optimises decision-making in real time.

## 1. Introduction

Falls remain a leading cause of serious injury, disability, and mortality among older people. Although multiple fall detection systems have been developed in recent decades, significant limitations persist in terms of their reliability, accuracy, cost, and applicability in real-world environments. In addition, the lack of comprehensive and up-to-date comparative reviews makes it difficult to identify more appropriate solutions tailored to the specific needs of this population. This situation limits the adoption of more effective technologies that help prevent adverse consequences and optimise emergency response.

Age is a relevant factor for the increased risk of falls, as shown in Figure 1 [1].

Various studies indicate that older women have certain protective factors against serious falls, mainly derived from more constant self-care habits, a greater willingness to seek healthcare and participate in preventive programs, as well as the adaptation of behaviours and environments that reduce exposure to risks. On the contrary, older men tend to suffer falls at younger ages and show a higher mortality associated with these events, a situation linked to the accumulation of comorbidities, the delay in seeking medical attention, the persistence of high-risk activities despite physical limitations, and the lower consideration of pathologies such as osteoporosis. This increases the severity of the injuries sustained [1].

Research and development of fall detection systems is essential to improve the quality of life of the elderly population, a demographic group that is constantly growing and particularly vulnerable to falls with potentially serious consequences. The incorporation of advanced technologies allows the creation of real-time, accurate and efficient monitoring systems, which not only enable an immediate response but also offer security to users and their caregivers. In addition, these systems allow the analysis of movement patterns, facilitating the prediction of risks and promoting active, autonomous and independent ageing [2].

Several studies have compared fall detection systems based on inertial sensors, evaluating their performance against those that exclusively use accelerometers [3]. The latter stand out for their speed and precision in registering sudden changes in acceleration in multiple directions [4,5]. Wearable devices, such as wearable bracelets, necklaces, or sensors, offer convenience and flexibility, allowing continuous monitoring without relying on home infrastructure [6,7]. In contrast, non-portable systems installed in the environment (such as cameras or pressure sensors) have a scope limited to specific areas [8,9]. The ability of wearable devices to integrate into daily life is considered key to fostering independence in older adults, reducing the feeling of dependency associated with the use of home-installed systems [10,11]. Likewise, its effectiveness in terms of usability and comfort has been analysed [12], highlighting its usefulness during various daily activities, from walking to physical exercise (30) [10]. However, their acceptance can be affected by the perception of intrusiveness, which highlights the importance of developing solutions that are not only efficient in detection but also comfortable and aesthetically appealing [9,13].

Figure 2 shows a bibliometric graph of the occurrence of terms from 913 studies of the SCOPUS database that relate to the detection of falls in elderly adults. To identify these articles, the following search string was used: (“fall detection”, OR “fall monitoring”, OR “fall prevention”, OR “fall alert”), AND (“elderly” OR “aged”, OR “senior”, OR “older adult”), AND (“system” OR “technology” OR “device”, OR “solution”), AND (“sensor” OR “wearable”, OR “camera”, OR “algorithm”), AND (“healthcare” OR “safety” OR “well-being”, OR “assistance”).

The co-occurrence graph illustrates the relationship between key terms extracted from the scientific literature on fall detection systems in older adults. Two major thematic groupings are clearly distinguished: on the one hand, a technological cluster, represented in red tones, concentrates concepts such as fall detection, wearable sensors, machine learning, deep learning and the Internet of Things, highlighting the importance of wearable sensors, artificial intelligence algorithms and machine learning to optimise the accuracy of systems of monitoring. On the other hand, the green cluster groups terms linked to clinical and demographic factors such as humans, aged, male, female, accidental falls and fall prevention, which reflects the concern for prevention, the analysis of gender and age variables, and quality of life. The dense network of connections between the two nuclei evidences the growing integration between technological innovations and care and prevention approaches, underlining the need for interdisciplinary solutions that respond to the demands of an ageing and vulnerable population.

Recent technological advances have driven the development of contextual fall detection systems, which integrate multiple data sources (such as sound, video, and accelerometry) to improve accuracy and reduce false positives [13]. However, these systems still face limitations related to their cost and accessibility.

Solutions based on machine learning and artificial intelligence algorithms have been implemented to interpret sensory data more effectively [4,14]. These algorithms allow detection to be adapted to different movement patterns, significantly improving accuracy [15,16,17]. Some models also analyse the user’s movement history, which allows falls to be identified with greater certainty [17]. This approach is key to overcoming the limitations of traditional systems based solely on fixed thresholds, especially in individuals with varying physical characteristics [9,18].

The growing use of cameras and real-time monitoring systems has raised concerns about the privacy and security of personal data. While environmental sensors such as cameras and microphones increase accuracy, they also present ethical and legal challenges, especially in the case of older people [19,20]. Therefore, developers must prioritise the protection of personal information in their designs [5].

The advancement of hybrid solutions that combine wearable devices with environmental monitoring systems has been evidenced. These approaches not only improve diagnostic accuracy but also promote sustainable design focused on user autonomy, such as in systems that integrate accelerometers into wearable devices with sensors strategically placed in the home [9,19]. This integration enables the identification of risk situations based on previous movement patterns and multi-source data [21,22].

The study and development of fall detection and monitoring systems for older people directly contribute to several United Nations Sustainable Development Goals (SDGs) (Figure 3), positively impacting their health and well-being and helping to prevent serious fall-related injuries and ensuring that they receive timely care in the event of an incident (SDG 3). The creation of innovative technology platforms and integrated systems contributes to the development of sustainable and resilient infrastructures. The integration of wearable devices with environmental monitoring fosters innovation in healthcare (SDG 9). By providing accessible and effective monitoring solutions for older adults, it contributes to reducing inequalities in healthcare and support for vulnerable populations, accessible regardless of the user’s socioeconomic status (SDG 10). These systems used in urban and community settings contribute to the creation of safer spaces for older adults by promoting inclusive and sustainable communities where citizens can live healthily and actively (SDG 11). Collaboration between different sectors, including health, technology and public administration, is essential for the development and implementation of these systems (SDG 17).

This work synthesises the last 15 years of development in fall detection systems for older adults. Knowing the scope of these developments provides a foundation for the design of new solutions with similar objectives carried out in academy and technology centres. The information provided can be helpful for designers of technologies for health and improvement of the quality of life that promote active and healthy ageing. A greater use of this type of solution makes it possible to reduce the demand for medical care in health homes by providing predictive solutions.

Section 2 of this document presents the methodology based on the PRISMA^®^. Section 3 presents the findings in scientific literature on multiple systems and technologies. Finally, Section 4 discusses crucial areas for improving the effectiveness of the systems developed and proposed for the future. Finally, the conclusions are presented.

## 2. Methodology

This scoping review was carried out following the guidelines of the PRISMA^®^ methodology. The dataset with the details of the review is available at [23]. Scientific articles published in the last 15 years, obtained from databases such as SCOPUS, Science Direct, Web of Science, PubMed, IEEE xplore and Taylor & Francis, have been considered.

Table A1 of Appendix A indicates the pages where relevant information of the sections of this document can be found. The systematic review consisted of three phases: formulation of research questions, delimitation of scope and exhaustive search for reference documents.

The main research question was as follows: What technological developments have been developed in the last 15 years for the detection of falls in older people, and how have they improved the quality of life of users? This question is relevant since early detection of falls can prevent serious injuries and improve the quality of life of the older adults, who are particularly vulnerable to this type of incident. The first objective of this research was to synthesise information on systems developed for the detection of falls in older adults and to compare their performance and operational characteristics. As a second objective, it was proposed to identify the sensor technologies used in fall detection systems. As a third objective, functional characteristics were compared, and trends in the development of these technologies were identified. In addition to this, ethical considerations were highlighted that addressed the use of these technologies, such as user privacy, the management of sensitive data and the need to obtain informed consent for monitoring.

The research questions that were raised for the extraction of information in the reference documentation were as follows: RQ1. What technologies have been developed in the last 15 years for the detection of falls in older adults? RQ2. What operating principles do technologies use to detect falls in older adults? RQ3. What advantages and disadvantages have been identified in the use of these technologies? RQ4. What prospects for the future are mentioned about the development of these technologies? RQ5. What challenges and limitations are addressed around the use of these technologies? The quality of the scientific articles was evaluated with the criteria described in Table 1.

Figure 4 shows the workflow for the selection of reference documents according to the guidelines of the PRISMA^®^ methodology.

The search for reference documents was performed using the published scientific literature databases SCOPUS, Science Direct, Web of Science, PubMed, IEEE explore and Taylor & Francis. Two researchers selected and included or excluded articles published in the last 15 years following PRISMA^®^ methodological guidelines with the keywords and Boolean operators explained in Table 2, determining a Cohen’s Kappa coefficient of 0.5489 (moderate agreement). Four steps were performed to identify articles, shown in Figure 4: (1) relevant articles were identified through a database search; (2) the articles were selected from their abstracts; (3) the full texts were reviewed and evaluated; and (4) eligibility decisions were made for these items.

The search for scientific articles was carried out using the keywords, “fall detection system elderly people”, specifying the search by abstract, title, and full text, and considering publications from the last ten years, including only peer-reviewed articles in English (Figure 2). A total of 82 duplicate documents were excluded from the total of 257 articles identified. Then, 47 studies were excluded according to the abstract, leaving 128 articles. A total of 54 documents that did not technically address the multiple technologies of interest were eliminated, leaving 74 articles, of which 70 reference articles could be accessed.

### 2.1. Inclusion Criteria

Studies describing the operation and tests carried out in the field have been preferred. In addition, priority has been given to the selection of articles published in repositories and scientific databases such as SCOPUS, Science Direct, Web of Science, PubMed, IEEE Xplore, and Taylor & Francis. Systematic reviews and articles of evaluations of fall detection systems for adults were considered as reference documents. Articles considered with these criteria ensure that research is considered that not only presents theories, but also practical results and is applied in real environments, guaranteeing the relevance and applicability of the findings in the field of elderly health and safety. Table 2 describes the text strings used in searching for related articles.

### 2.2. Exclusion Criteria

Studies that evaluate control algorithms or computational cost performance, but do not focus on the practical applications of fall detection and monitoring technologies, have been excluded. This includes research that is limited to theoretical studies without a practical component or that does not demonstrate results in real environments, thus ensuring that only works that provide relevant and applicable information to the field of elderly health and safety are considered.

## 3. Results

The findings in the scientific literature are classified in this section, addressing developments in fall detection systems for older adults, the principles of operation, the challenges of their implementation, future perspectives, and the contributions of artificial intelligence.

### 3.1. Developments of Fall Detection Systems in Older Adults

The scientific literature has presented a large number of developments of fall detection systems for older adults. These can be classified into main categories based on the technology of communication and the measurement of data through sensors. They have been categorised (Figure 5) into Inertial Sensors, Computer Vision, Pressure, Vibration, and Sound Sensors, Internet of Things (IoT), Wearables, and Machine Learning. Although combinations of these technologies exist, Figure 5 is constructed to facilitate the reader’s identification of these developments based on their names and most representative technologies.

#### 3.1.1. Inertial Sensors

Fall detection systems have been developed and evolved through the use of inertial sensors, such as accelerometers and gyroscopes, with the aim of improving accuracy and reliability in the identification of these events [24]. Systems based on three-dimensional accelerometers and classification algorithms have been developed, such as fall detection by feature extraction and support vector machines (SVMs), and wearable devices with three-dimensional accelerometers [13]. Other approaches have incorporated IoT and Big Data to optimise detection in older people [18], while some studies have proposed mobile activity recognition systems based on the Ameva algorithm [12]. Improvements in fall detection using triaxial accelerometers have also been designed [25], although challenges remain in autonomous systems based on wrist sensors [26]. To address the need for adaptive solutions, custom systems such as Chameleon have been developed, which adjust fall detection to each user in home environments [27]. On the other hand, methods that combine accelerometers and gyroscopes in wearable devices have been implemented, such as fall detection based on gyroscopic sensors and accelerometers in smartphones [6], the use of SVM in low-cost Android devices [12], and the integration of detection algorithms into smart home systems using EnOcean devices [28]. In addition, wireless sensor networks for fall detection [5] and high-precision systems based on smartphone inertial sensors [29] have been developed, demonstrating the potential of these devices to improve the safety of older people.

#### 3.1.2. Machine Vision

Fall detection systems based on computer vision have evolved through the use of cameras and image processing to analyse the visual characteristics of human silhouettes and their postures. Approaches employing video for fall detection in smart home environments [4], spatiotemporal feature-based recognition [7], and camera surveillance have been developed to improve the safety of older people [10]. Detection systems based on the variation of key points of the human skeleton [30], on the analysis of integral images and the histogram of oriented gradients [20], as well as on the use of the gravity vector with wide-angle cameras [31], have also been proposed. Other methods have explored 3D image analysis for indoor detection [32], the use of machine vision techniques in RGB-D imaging [22], and positive semi-defined matrix-based detection [33]. Additionally, Kinect sensors have enabled fall detection using real-time spatial data, with applications in motion monitoring platforms [16], mobile assistive robots [34], and in-depth 3D image analysis [32]. The application of neural networks has driven intelligent video-based systems, such as fall detection with deep learning models [35] and the use of transfer learning in combination with IoT and thermal image-assisted surveillance [36], improving accuracy and efficiency in identifying fall events in older people.

#### 3.1.3. Pressure, Vibration and Sound Sensors

Fall detection systems based on pressure, vibration and sound sensors have explored various strategies to improve the accuracy in the identification of these events. Methods have been developed that employ microphones and ground vibration analysis to detect falls and estimate gait in older people living alone [15]. In addition, the use of capacitive sensors integrated into the ground has enabled the creation of intelligent measurement systems for fall detection with greater accuracy [37]. On the other hand, the analysis of environmental sounds has proven to be an effective tool in the identification of falls, using ternary local acoustic patterns to process and classify sound events associated with these incidents [38]. These approaches complement other detection systems, providing viable alternatives for non-intrusive monitoring and improving the safety of older people in different settings.

#### 3.1.4. Internet of Things

Internet of Things (IoT)-based fall detection systems and sensor networks have enabled data transmission to the cloud and connected devices, improving responsiveness and real-time monitoring. Approaches that combine mobile devices with IoT and machine learning, both in the cloud and at the network edge, have been developed for fall detection in older adults [39]. Energy-efficient Bluetooth-based systems for fall detection and warning have also been implemented in nursing homes [40], as well as wearable sensor nodes optimised to reduce energy consumption in IoT fall detection systems [41]. In addition, the integration of multiple sensors into smart environments has led to the development of advanced platforms that not only detect falls, but also assess users’ social and contextual risk to offer a more personalised and efficient response [42]. These advances consolidate the IoT as a key tool in improving the safety and well-being of older adults.

#### 3.1.5. Portable Devices

Fall detection systems based on wearable devices and wearable devices have advanced by integrating sensors into clothing and accessories to offer discreet and efficient solutions. An innovative approach has been the development of bright socks specifically designed for fall detection in older women, using a user-centred design to improve comfort and effectiveness [5]. In addition, smartwatches and wristbands have proven to be practical tools for continuous monitoring, with detection methods based on data from accelerometers embedded in wrist-worn devices [43]. The use of triaxial accelerometers on the wrist has improved the accuracy of identifying falls in older people, facilitating the implementation of early warning systems [44]. These advances in wearable devices reinforce the importance of wearable solutions in fall prevention and detection, providing an accessible and easy-to-use alternative for health monitoring in older adults.

#### 3.1.6. Machine Learning

Fall detection systems have incorporated machine learning models and neural networks to improve accuracy and adaptability in identifying these events. Classification algorithms such as support vector machines (SVMs), decision trees, and neural networks have been implemented in the detection of falls in older people [17]. In this context, deep learning-based models capable of classifying falls from multiple datasets [45] have been proposed, as well as improved approaches through feature weighting and Bayesian classification [46]. Another significant advance has been the combination of neural networks with low-power wireless sensor networks, which has enabled accurate detection and location of falls in real-time [47]. These developments highlight the potential of machine learning to optimise fall detection, facilitating the implementation of intelligent and adaptive systems for the safety of older adults.

#### 3.1.7. Advances in Early Fall Detection Systems for Older Adults

Early fall detection systems for older adults have increasingly focused on the pre-impact phase, that is, the period prior to body contact with the ground, with the objective of activating countermeasures and preventing severe injuries. In this context, wearable inertial sensors (accelerometers and gyroscopes) are strategically placed on specific body locations—such as the waist—to capture accelerations, angular velocities, and inclination patterns that precede a fall. Recent studies have achieved anticipation intervals of approximately 400 ms using threshold-based algorithms [48], as well as hybrid deep learning models combining convolutional and recurrent architectures (CNN-LSTM) [49]. More recently, deep residual network models designed for pre-impact detection have attained an accuracy of 91.87% for falls and 92.52% for impacts using the KFall dataset [50].

The integration of machine learning algorithms, such as Support Vector Machines (SVM) and Random Forests, together with multisensor fusion (accelerometer + gyroscope + electromyography), has enabled improvements in sensitivity and reductions in false alarms in pre-impact detection. A fractional-domain technique has also been successfully applied in older adult populations to anticipate falls [51]. Despite these advances, several challenges remain, including the need for optimal sensor placement, validation in real-world environments with older adults, and the standardisation of evaluation protocols to enable comparative system assessment.

### 3.2. Technical and Functional Aspects of Fall Detection Systems

Table 3 presents functional details of each system according to the published studies, sensitivity/recall, accuracy, specificity, robustness autonomy, portability have been considered and the systems have been tabulated grouped according to their classes inertial sensors, based on computer vision, based on pressure, vibration and sound sensors, based on the internet of things (IoT) and sensor networks, based on portable devices and wearable devices, and based on machine learning models and neural networks.

The information presented in Table 3 is organised according to the manner in which it is addressed in the studies identified. Although similar variants exist across several types, it was deemed appropriate to group them based on the titles of the works included in the scientific literature reviewed.

Figure 6 shows a comparative graph of the parameters of each of the systems in Table 3. The parameters that are graphed are accuracy, sensitivity, specificity, robustness and autonomy. Sensitivity or recall measures the ability to correctly detect positive cases, while accuracy reflects the overall percentage of correct predictions. Specificity focuses on properly identifying negative cases, thereby reducing false positives. Meanwhile, robustness assesses how the system responds to errors or adverse conditions, autonomy indicates its independence from human intervention, and portability refers to the ease with which it can be transferred and applied across different environments [10,11].

Figure 6 highlights multiple developments around values greater than 90% accuracy in detection and sensitivity. About 50% of developments are considered robust and mostly of high specificity. Medium and high autonomy systems stand out. Most of these devices have high portability depending on the technologies used.

### 3.3. Key Technologies of Fall Detection Systems in Older Adults

The multiple technologies that have been used have particular characteristics and will be addressed in the following titles by their operating principles, their advantages, challenges and technological innovations.

#### 3.3.1. Inertial Sensors for Fall Detection Systems of Elderly People

Inertial sensors, particularly accelerometers and gyroscopes, have become integral components of fall detection systems for older people. These sensors are integrated into wearable devices to monitor body movements and detect falls (Figure 7), thus improving the safety and well-being of older people [63].

Systems based on inertial sensors have proven to achieve high accuracy in fall detection. For example, one study reported an accuracy of 95.67%, with a sensitivity of 99.0% and a specificity of 95.0% [29]. Another system achieved an accuracy of 95.44% [64].

These systems allow for real-time monitoring and immediate alerting to caregivers, which is crucial to provide timely assistance and reduce the severity of injuries [29,64].

Some advanced systems are capable of detecting falls before impact, making it possible to activate preventive measures. These systems have demonstrated high sensitivity and specificity, being able to detect falls up to 0.1875 s before impact [47,65].

#### 3.3.2. Fall Detection System for Elderly People Based on Computer Vision

Vision-based systems employ computer vision and artificial intelligence technologies to monitor and detect falls without requiring portable devices, which are often forgotten or rejected by older users [66,67]. These systems use different types of data to capture the environment and analyse body movement (Figure 8), such as RGB (red, green, blue), depth, and infrared (IR) images [33].

Some alternatives operate with this type of technology, some for object detection and tracking, shape and motion analysis, deep learning algorithms, multimodal learning AI at the edge (Edge AI), and human dynamic stability descriptors. Object detection and tracking employ algorithms such as YOLO (You Only Look Once) and Kalman filters to identify and track human movements [33,66].

For the analysis of shape and movement, techniques such as elliptical adjustment, optical flow, and morphological variation analysis are applied to detect falls through changes in posture and body displacement [36,44,68].

Machine learning algorithms employing models such as support vector machines (SVMs), LSTM networks, and deep learning models are used to classify movements and detect falls more accurately [33,69]. There are more advanced developments that incorporate multimodal learning, which combine environmental and physiological signals to improve detection accuracy [40]. Developments such as AI at the Edge (Edge AI) are made up of embedded systems such as NVIDIA Jetson Nano, which make real-time processing possible for immediate detection [27]. Human dynamic stability descriptors prioritise stability and balance indicators over kinematic variables to detect actual falls [70].

#### 3.3.3. Fall Detection Systems Based on Pressure, Vibration and Sound Sensors

Various technologies based on pressure, vibration and sound sensors (Figure 9) have been developed to address this problem with varying levels of accuracy, sensitivity and specificity [71].

##### Pressure Sensor-Based Systems

Some developments based on these technologies include shoe insoles with pressure sensors and smart mats with piezoresistive sensors [72]. Portable devices equipped with film-type pressure sensors, integrated into templates, have been designed. These sensors collect information on plantar pressure distribution, which is analysed by threshold algorithms and support vector machines (SVMs) to detect falls. This method has demonstrated high reliability and accuracy in experimental environments [73,74].

Another innovative technique incorporates differential piezoresistive sensors embedded in smart carpets. This system has reported a sensitivity of 88.8% and a specificity of 94.9%, making it suitable for home care settings [75].

##### Systems Based on Vibration Sensors

Vibration sensors for floors and systems that combine vibration and sound in controlled environments have been developed with this technology.

Floor vibration systems use floor-mounted sensors, such as microphones, to pick up vibrations generated by movements or falls. By analysing the frequency characteristics of the recorded signals, it is possible to identify dips and gait states with good results in experimental environments [26]. A more advanced approach combines vibration sensors on the floor with acoustic detection, using signal processing techniques and pattern recognition algorithms. This system achieved a sensitivity of 97.5% and a specificity of 98.6% in simulated fall detection [76].

##### Sound Sensor-Based Systems

Acoustic sensors can pick up the sound pressure generated by a fall. However, the isolated use of this signal is not reliable for accurate detection [77]. The combination of sound sensors with accelerometers, along with the use of algorithms based on fuzzy logic, has been shown to significantly improve accuracy by reducing false alarms [78,79]. Some systems use multiple microphones to simultaneously analyse the source of sound and ground vibrations, allowing for more accurate estimation of driving conditions and detection of falls [26].

An important variant of these methods is the fall detection system based on millimetre-wave (mmWave) radar, which has emerged as a non-intrusive solution with high spatial resolution for monitoring older adults in clinical and home environments, as it avoids image recording and therefore protects user privacy [80,81]. This technology leverages FMCW signals and micro-Doppler signature analysis to recognise kinematic patterns associated with a fall, employing classification models based on deep neural networks such as LSTM, CNN, or optimised hybrid architectures for real-time performance. Recent studies report high accuracy and robustness to inter-subject variability, integrating techniques such as wavelet transform, point-cloud voxelization, and contextual gait recognition, as well as hardware-accelerated implementations to ensure low latency. Representative system evaluations demonstrate accuracy levels exceeding 98%, positioning mmWave as a mature, scalable, and privacy-preserving alternative by design [82,83].

In summary, we can conclude that pressure sensors have proven to be effective when integrated into smart shoe insoles or carpets, as they allow accurate and reliable detection of falls. Vibration sensors, installed on the ground, help identify both falls and gait patterns by analysing the frequencies generated by movement. As for sound sensors, their performance is significantly improved when combined with other devices, such as accelerometers, and by employing advanced algorithms—such as fuzzy logic—that reduce false alarms and increase the accuracy of the detection system. Table 4 summarises the most important findings of these types of technologies.

#### 3.3.4. Fall Detection Systems Based on IoT and Sensor Networks

These solutions integrate multiple sensory technologies, data analytics, and machine learning algorithms for accurate and efficient fall detection (Figure 10). The technologies that make up these solutions include sensors, machine learning, deep learning, and IoT architecture [84].

IoT-based fall detection systems integrate various types of sensors to improve their effectiveness. Wearable sensors, such as accelerometers, gyroscopes, and heart rate sensors, are widely used to monitor body movement and detect falls [85,86,87]. On the other hand, environmental sensors, including depth cameras, atmospheric pressure sensors, vibration sensors, and acoustic sensors, enable the monitoring of the environment to identify fall events [88,89]. Merging data from multiple sensors has proven to be an effective strategy to increase system accuracy and reduce the rate of false alarms [89,90].

Machine learning and deep learning employ algorithms, assembly methods, and generative adversarial networks (GANs). The algorithms employ convolutional neural networks (CNNs), LSTM networks, and recurrent neural networks (RNNs) to analyse sensory data and detect falls with high accuracy [91]. Assembly methods employ assembly-based algorithms, such as stacking, to increase prediction accuracy and system robustness [92]. Generative adversarial networks (GANs) are applied for semi-supervised learning, addressing data imbalance issues and improving system performance [86].

IoT architecture is based on mainly wireless communication technologies (WiFi, Lora, ZigBee, BLE, GSM, LTE/4G, 5G…) and on the distribution of computing at different points of the architecture (edge computing, fox computing, cloud computing) [93,94]. Edge and Cloud computing enable rapid data processing and management, ensuring timely detection and effective response [95].

Alert and notification systems generate notifications to caregivers or emergency contacts through mobile applications, emails or SMS when a fall is detected [96]. Contextual responses consider the availability and location of caregivers to ensure adequate care [97]. Performance evaluation considers accuracy, sensitivity, and real-time monitoring. High accuracy and sensitivity are critical for effective detection. Systems employing advanced algorithms and sensor fusion have reported accuracy levels above 95% [86,90]. Continuous, real-time analysis of data is essential to intervene quickly and minimise the adverse consequences of falls [98].

#### 3.3.5. Wearable Devices for Fall Detection Systems

There are multiple types of wearable devices, from phones and smartwatches to activity trackers, smart glasses, and medical monitoring systems (Figure 11) [99]. Each has specific functionalities and benefits.

Smartwatches and activity bracelets are widely used for health monitoring, physical tracking, and receiving notifications [100,101,102]. Smart glasses are used in augmented reality applications and to access data over Wi-Fi networks [103]. Medical wearables include devices designed to monitor vital signs, assist in medical diagnoses, and provide care to older people [104,105].

These technologies employ more than sensors, communication modules, and microcontrollers. The sensors often employed are accelerometers, gyroscopes, magnetometers, and biosensors to collect data in real-time [105,106]. Communication modules use technologies such as Bluetooth, ZigBee, RFID, and NFC for wireless data transmission [96]. Advanced microcontrollers, combined with AI algorithms, enable the processing of collected data, generating useful information and automating tasks [104,105,107].

Wearable devices have a wide range of applications in various sectors. In the field of health and well-being, they are used to monitor physiological parameters, support medical diagnoses, and promote healthy lifestyles [101,103,104]. In industry, they contribute to improving the performance and safety of personnel through continuous monitoring [94]. In the field of sports, they optimise the user experience during training and professional competitions [104]. Finally, in virtual environments, they allow immersive navigation and interaction in simulated realities [108]. Table 5 presents some developments of wearable devices that use IoT technology.

#### 3.3.6. Fall Detection Systems Using Machine Learning and Neural Networks

Recent advances in machine learning (ML) and neural networks have dramatically improved the accuracy and efficiency of these systems.

Machine learning (ML) models typically use data from wearable devices equipped with inertial measurement units (IMUs) and vital signs sensors (Figure 12). The combination of these data allows for improved accuracy in fall detection [99,109,110]. For example, the joint use of IMU and heart rate sensors has been shown to improve the performance of classifiers [110].

Lightweight models, such as TinyML, have been designed for resource-constrained environments, including wearable devices, and are characterised by achieving high accuracy with low computational power, making them particularly suitable for embedded systems [111]. In addition, several machine learning (ML) algorithms for fall detection have been evaluated, including k-Nearest Neighbours (KNN), Support Vector Machines (SVM), Random Forest, and Naïve Bayes, which have demonstrated a high level of accuracy in controlled environments. However, they have limitations when implemented in real-time scenarios [112]. Convolutional neural networks (CNNs) have been employed in fall detection due to their ability to automatically extract features from sensory data, demonstrating high effectiveness in differentiating falls from everyday activities [112,113]. Developments with CNN have achieved accuracies of around 99.29% in wearable systems. Advanced models, such as ResNeXt and dual-flow convolutional neural networks (DSCN), have further improved detection using complex feature extraction mechanisms and automatic attention [113,114,115], achieving accuracy rates greater than 99% in public datasets. The combination of different architectures, such as CNNs and recurrent units with gates (GRUs), has enabled the capture of spatial and temporal dependencies in sensory data, thereby improving classification performance [116].

As for the sensors used, the notable ones include accelerometers, gyroscopes, sound signals, and infrared sensors. Accelerometers and gyroscopes have been used in wearable devices, providing essential data on movement [114,117,118]. Both ML and deep learning algorithms have employed acoustic cues for fall detection, achieving high accuracy by analysing audio features [119]. The application of deep neural networks to data obtained by IR sensors has shown significant improvements in detection accuracy [120].

### 3.4. Advantages and Disadvantages of Fall Detection Systems

The variety of solutions described in the previous sections has multiple common characteristics. Table 6 shows the most representative advantages and disadvantages according to each type of technology.

#### 3.4.1. Advantages and Disadvantages of Inertial Sensor-Based Fall Detection Systems for Adults

Fall detection systems based on inertial sensors, especially those that use accelerometers and gyroscopes, stand out for their high precision, reaching accuracy rates of up to 99.52%, with sensitivity and specificity greater than 98% [64,121]. This accuracy makes it possible to effectively differentiate between real falls and everyday activities, reducing the number of false alarms [29]. In addition, they offer real-time detection and immediate alerting to caregivers or emergency services, which is vital for timely medical intervention [122]. Another advantage is their portability and convenience, as these sensors are lightweight and can be integrated into devices such as belts, bracelets or even electronic textiles, which favours their continuous use without interfering with the user’s routine [123]. Finally, by not requiring video recording, these systems safeguard the user’s privacy, a considerable advantage over vision-based systems [124].

Despite their high accuracy, these systems still have limitations, such as false alarms and omissions in detection, which can decrease user confidence and lead to rejection of their use [125]. Differentiating between a fall and a similar action, such as sitting abruptly, remains a relevant technical challenge. In addition, the need for continuous monitoring leads to high power consumption, which compromises battery life in portable devices [126]. Real-time data processing, particularly when employing advanced machine learning algorithms, requires significant computational resources, which may constitute a limitation for devices with low processing capacity and limited data storage capability [86]. Finally, the effectiveness of the system depends to a large extent on the correct placement and constant use of the sensors, something that can be difficult for users with cognitive impairment. In addition, installation and maintenance of the system may require specialised technical knowledge [127].

#### 3.4.2. Advantages and Disadvantages of Computer Vision-Based Adult Fall Detection Systems

Fall detection systems based on computer vision have essential benefits, starting with their non-invasive nature, since they do not require users to wear physical sensors, which increases comfort and acceptance, especially among older people [128]. Likewise, advanced models such as YOLOv5s and BlazePose have demonstrated high accuracy and efficiency, enabling real-time detection with highly reliable results [129]. Their implementation is relatively simple, as they can use inexpensive RGB cameras and do not require complex infrastructures, which favours their accessibility [130]. In addition, these systems can be easily integrated with IoT technologies, making it easier to continuously monitor and send automatic notifications in the event of falls [131]. Finally, they are robust in various scenarios, as they can operate in different environmental and lighting conditions, increasing their versatility in multiple contexts [69].

Despite their many benefits, machine vision systems face significant challenges, most notably privacy concerns, as their operation requires continuous image or video capture of the environment and the people being monitored [132]. In addition, these systems often demand high processing power, which makes them challenging to implement in devices with limited computational resources [129]. Their performance can also be compromised by unfavourable environmental conditions, such as poor lighting, obstructions in the field of view, or unsuitable camera angles [133]. Another critical aspect is the difficulty of generalisation, since variations in the physical appearance of users or in environments can affect the accuracy of the model in situations not contemplated during training [133]. Finally, although cameras may be low-cost, the initial investment in specialised hardware and analysis software can represent a considerable expense [69].

#### 3.4.3. Advantages and Disadvantages of Fall Detection Systems in Older Adults Based on Pressure, Vibration, and Sound Sensors

Fall detection systems based on pressure, vibration, and sound sensors offer high accuracy, with studies reporting sensitivities of 97.5% and specificities of 98.6% when combining vibration and sound signals [134]. In addition, the integration of pressure and sound sensors makes it possible to improve detection accuracy further, significantly reducing false alarms [135]. One of their main advantages is that they are non-intrusive, as they do not require the user to carry devices, which is ideal for older people who might find body sensors uncomfortable [125]. These systems can be passively integrated into the environment, such as in smart carpets or sensorised floors, making them easy to implement without interfering with everyday activities. In addition, triboelectric sensors and other similar systems are low-cost and easy to install, making them suitable for domestic and institutional environments [127]. Finally, they allow continuous and real-time monitoring, generating immediate alerts upon the detection of falls [136].

Despite their effectiveness, these systems can generate false alarms due to the difficulty of differentiating between real falls and similar everyday activities, such as dropping an object or moving abruptly [135]. In addition, its accuracy can be compromised by external factors such as ambient noise or vibrations unrelated to falls [134]. Another major challenge is the location of the sensors: their effectiveness depends on a correct distribution in the environment, which may require careful planning to cover all critical risk areas [137]. In terms of energy consumption, some systems have a high demand, which can make it challenging to operate continuously without interruptions [76]. Finally, while specific sensors are easy to install, others require more complex configurations and regular maintenance, which could pose an obstacle to their widespread deployment [127]. Additionally, it may be noted that users show reluctance to install these devices in their homes, as well as to allow the entry of technicians who regularly perform the maintenance of such equipment.

#### 3.4.4. Advantages and Disadvantages of IoT-Based Fall Detection Systems and Sensor Networks

Fall detection systems based on IoT devices employing motion sensors, such as accelerometers and gyroscopes, offer high accuracy in identifying fall events, especially when integrated with machine learning algorithms and deep neural networks, which improve accuracy and reduce false alarms [138]. In addition, the use of IoT technologies allows for real-time monitoring and immediate data transmission, facilitating rapid responses to emergencies [89]. Edge and cloud computing bolster this capability, increasing system efficiency. Another key advantage is respect for user privacy, as technologies such as infrared sensors, thermal cameras, and WiFi-based systems enable accurate detection without requiring wearable or invasive devices [139]. These systems also stand out for their adaptability, allowing customisation according to the environment and specific conditions through online calibration algorithms and continuous learning. In addition, integration with voice assistants and cameras enables contextualised response and post-incident assessment [92].

Despite their high performance, these systems may face accuracy issues under certain conditions. For example, wearable sensors may have difficulty distinguishing between actual falls and similar movements such as lying down, while machine vision systems are affected by low-light conditions or when the user is out of the visual field [128]. In terms of usability, wearable devices can be uncomfortable, especially for older adults, and environmental sensors limit monitoring to specific spaces, restricting their usefulness in everyday life. In addition, the implementation of IoT-based solutions and sensor networks entails high costs and technical complexity, including the need for infrastructure, maintenance, and constant connectivity, which can be a challenge in areas with limited internet access [140]. Finally, the transmission of personal data over IoT networks poses security risks that require robust protection systems, as well as ethical considerations regarding privacy and the handling of sensitive user information [139].

#### 3.4.5. Advantages and Disadvantages of Fall Detection Systems Based on Wearable Devices

Wearable devices such as smartphones, smartwatches, and specialised vests offer remarkable portability and ubiquity, allowing continuous monitoring regardless of the user’s location, unlike static sensors limited to specific areas [141]. These devices, being lightweight and integrable into everyday clothing, are less intrusive and more comfortable for daily use [142]. In addition, combining multiple sensors (such as accelerometers and gyroscopes) with data fusion techniques, e.g., combining camera data with inertial sensors, improves detection accuracy and reduces false alarms [143]. Advanced algorithms such as those based on fuzzy logic and machine learning further strengthen the system’s ability to distinguish between actual crashes and normal activities [144]. Another key benefit is cost-effectiveness, as it leverages widely available personal devices, such as smartphones, which already include embedded sensors, reducing the need for additional hardware [145]. Finally, these devices offer real-time detection and immediate alerts, which can significantly reduce the time to medical assistance [146].

One of the main limitations of wearable devices is the reliance on user cooperation, as they must be used consistently, which can be uncomfortable or impractical, especially for older people [147]. There is also a risk that the user will forget to put on the device or decide to remove it, thus compromising the effectiveness of the system [148]. Another critical aspect is battery life: real-time monitoring and data transmission consume power continuously, forcing frequent recharges and constant maintenance, a task that can be difficult for older adults [145]. Despite technological advances, false alarms are still present due to misinterpretations of everyday movements, and detection may fail if the device is not correctly positioned or used inconsistently, as in the case of noise generated by hair or the relative movement of the upper limb when wearing a watch or wrist-mounted device [145]. Finally, when devices incorporate cameras or other means of visual monitoring, privacy-related concerns can arise, especially about continuous video recording and the security of sensitive health data [149].

#### 3.4.6. Advantages and Disadvantages of Fall Detection Systems for Older Adults Based on Machine Learning Models and Neural Networks

Fall detection systems based on machine learning models and neural networks have demonstrated high accuracy and sensitivity. In particular, convolutional neural networks (CNNs) achieve sensitivities of 99.05% and specificities of 99.68%, far outperforming threshold-based approaches [3]. Models such as support vector machines (SVMs) and k-neighbours have also reported accuracies close to 99% [4,5]. In addition, neural networks have a greater ability to detect real falls, avoiding false positives in similar events such as abrupt sitting, which improves user confidence [6]. The use of multimodal approaches and the fusion of data from multiple sensors (e.g., accelerometers and gyroscopes) further increases the accuracy of the system and reduces false alarms [7,8,9]. Artificial neural networks (ANNs), on the other hand, have a low computational cost, which makes it easy to implement them in portable devices that are comfortable for older people [10]. Finally, these systems allow for real-time fall detection and immediate sending of alerts using technologies such as Wi-Fi, ensuring prompt assistance [11].

Despite their advantages, these systems face several challenges. Collecting enough custom data to train models can be complex, which increases the rate of false positives if the system is not tuned correctly [12]. In addition, the phenomenon of model drift—where a model trained on one device performs poorly on another—raises scalability problems in portable systems [12]. Some machine learning models require extensive feature extraction and optimisation processes, which implies a high computational load and limits their applicability in resource-constrained devices [13]. In addition, the accuracy of these systems is highly dependent on the quality and correct placement of the sensors, which can vary depending on the environment or the type of device [14,15]. Another challenge is energy efficiency: while reducing the sampling rate saves energy, it can also compromise accurate detection [4]. Finally, in the case of vision-based systems, continuous monitoring raises significant concerns regarding privacy and personal data management, requiring powerful equipment and high-capacity data storage [16].

### 3.5. Challenges and Limitations

The challenges and limitations (C&L) of fall detection systems in older adults present a large number of dimensions according to the technologies they employ. Figure 13 shows, in a synthesised way, the most outstanding challenges and limitations.

#### 3.5.1. C&L in Inertial Sensor-Based Fall Detection Systems

One of the main challenges is the high rate of false alarms, which can generate frustration and decrease acceptance by older adults [127,150,151]. Strategies implemented to reduce this problem include post-fall behaviour analysis and the use of multimodal systems that combine data from different sensors [150,152]. Some older adults may show resistance to the use of wearable devices due to physical discomfort or cognitive impairment [127]. Therefore, it is essential to design comfortable and non-intrusive devices to promote their adoption [64].

#### 3.5.2. C&L in Fall Detection Systems in Adults Based on Machine Vision

Fall detection systems that rely on computer vision face several challenges and limitations that affect their reliability in real-world applications. One critical issue lies in physical and environmental variations, where factors such as camera positioning, occlusions, and complex backgrounds can significantly reduce detection accuracy. To address these limitations, researchers have explored the use of multi-view datasets and advanced tracking algorithms, which enhance robustness under diverse conditions [66,133].

Another persistent challenge is low-light conditions, which hinder image clarity and compromise system performance. Recent advances in image enhancement techniques, combined with deep learning-based tracking frameworks, have demonstrated potential to improve detection capabilities in poorly lit environments [153].

A further limitation concerns privacy issues, particularly in sensitive contexts such as elderly care. The deployment of depth cameras, as opposed to conventional RGB cameras, has emerged as a promising solution, as they enable accurate motion detection while preserving user anonymity [67].

#### 3.5.3. C&L in Fall Detection Systems Based on Pressure, Vibration and Sound Sensors

Fall detection systems that use pressure, vibration, and sound sensors have certain inherent limitations to their design and operation. First, their effectiveness is often restricted to specific environments, as they require fixed installation in areas where fall events are expected to occur, such as bedrooms, living rooms, or bathrooms. This spatial dependence limits their coverage, leaving those areas of the home unattended that are not equipped with the sensors [154,155].

In addition, this type of system requires physical installation within the living space, which may involve structural adaptations, wiring or integration with surfaces such as floors or walls. Although some devices are low-cost, their implementation may not be practical in all homes, especially those with architectural limitations or a small budget [156,157].

Another major limitation is that when used in isolation, these sensors do not offer complete reliability. As they are designed to pick up indirect signals (such as changes in ground pressure, sudden vibrations or impact sounds), they can generate a high rate of false alarms, activating in the event of non-dangerous movements or similar events that do not constitute real falls. For this reason, it is recommended to use it in combination with other types of sensors (e.g., inertial or visual), in order to increase the accuracy of the system and reduce incidences of erroneous notifications [158,159].

#### 3.5.4. C&L in IoT-Based Fall Detection Systems and Sensor Networks

The main challenges are privacy, data security and user comfort when using these devices. It is essential to ensure the secure collection and management of sensitive data generated by these systems [160]. Wearable devices should be comfortable and non-intrusive to ensure their continued use by older people [92].

#### 3.5.5. C&L in Fall Detection Systems Based on Portable Devices and Wearable Devices

The most critical challenges relate to security, data privacy, energy efficiency, and user acceptance. Protection of collected information is essential due to concerns about data theft, unwanted surveillance, and privacy preservation [102,161,162]. A balance between functionality and power consumption is required to avoid frequent loads that affect the user experience [106]. Ergonomic design and ease of use are determining factors for its mass adoption [105,163]

#### 3.5.6. C&L in Fall Detection Systems Based on Machine Learning Models and Neural Networks

Ensuring high accuracy in real-time applications remains a challenge. The implementation of continuous calibration and e-learning mechanisms is necessary to adapt to changing conditions [87,112]. In senior care settings, it is imperative to address security and privacy issues in the collection and use of data [164].

### 3.6. Future Directions

#### 3.6.1. Improvements Integration and Signal Processing

The development of fall detection systems has incorporated advanced data processing and analysis techniques that significantly enhance their accuracy and reliability. Notable among these are Kalman filters and machine learning classifiers, such as support vector machines (SVM) and artificial neural networks [29,42,65]. Likewise, the application of deep learning models, particularly convolutional neural networks (CNNs), enables differentiation between actual falls and routine daily activities, thereby improving detection accuracy [165,166].

In parallel, Internet of Things (IoT)-based systems integrate heterogeneous sensors—including Wi-Fi signals, floor pressure sensors, and GPS receivers—to provide comprehensive monitoring and strengthen user safety [127]. This technological convergence points toward interconnected ecosystems capable of delivering faster and more efficient responses.

Among emerging trends, there is a projected increase in the integration of computer vision systems with smart home environments, which will enable continuous monitoring and immediate responses to incidents [69]. However, one persistent challenge lies in the availability of representative datasets. The need for databases incorporating real falls in everyday contexts has been acknowledged as essential to optimise model training and improve overall effectiveness [167].

Future projections include the adoption of more sophisticated architectures, such as Mask R-CNN and Transformer-based models, which may further enhance detection efficiency and accuracy [33]. At the same time, simplifying user interfaces and integrating devices with home care services are anticipated as key strategies to promote adoption and effectiveness [88].

In the domain of portable and wearable devices, greater convergence with IoT is expected through the incorporation of embedded sensors, fostering the emergence of smarter and more interconnected ecosystems [103,168,169]. Additionally, the deployment of artificial intelligence algorithms will strengthen data analysis and enable the provision of personalised services [104,105]. Finally, the expansion of these systems is projected toward new domains, including personalised healthcare, interaction within virtual environments, and other emerging applications [108,170].

#### 3.6.2. Improvements in Systems Using AI

The future outlook for fall detection systems in older adults based on machine learning (ML) models and neural networks points toward a significant evolution in terms of accuracy, adaptability, and reliability in real-world environments. Although these models have demonstrated outstanding performance under controlled conditions, their application in real-life scenarios presents considerable challenges due to environmental variability and individual differences among users. Consequently, a priority line of research involves the development of systems capable of continuously refining and adapting through online learning techniques and self-adjusting models that respond to the dynamic behaviour of the user and the physical context in which they operate.

A substantial advancement is projected in the integration of multimodal data, combining information from inertial sensors, vital signs, environmental sensors, and even contextual data such as images or sound. This fusion of sources will enhance the robustness of the models, reduce false positives, and enable more precise and context-aware detection. In parallel, the development of context-aware systems, capable of interpreting situations beyond the physical event itself, will be crucial to effectively discriminate between an actual fall and similar activities, such as sitting abruptly or stumbling without losing balance.

A critical area accompanying this progress is data security and privacy. The increasing interconnection of these systems with IoT networks and wearable devices raises legitimate concerns regarding unauthorised access, unwanted surveillance, and the protection of sensitive information, particularly in medical and geriatric care contexts. Therefore, future research must balance system performance with the implementation of ethical frameworks and robust cybersecurity solutions. Figure 14 synthesises the aspects related to the aforementioned future directions.

## 4. Discussion

### 4.1. Technological Advances

Over the past decade, fall detection systems have seen remarkable progress thanks to the integration of emerging technologies such as high-precision inertial sensors, machine vision, environmental sensors (pressure, vibration, and sound), and artificial intelligence. In particular, the use of machine learning models and deep neural networks, such as convolutional networks (CNNs), has made it possible to achieve sensitivities of 99.05% and specificities of 99.68% [113]. These advances have not only improved the performance of systems but also allowed for more compact and efficient deployments for wearable devices [171].

One of the most promising approaches is the combination of multiple sensors such as accelerometers, gyroscopes, pressure and sound sensors, along with data fusion techniques. This integration has allowed for more robust and reliable detection. For example, merging data from cameras and inertial sensors has been shown to reduce false alarms [143,144]. Multimodal fusion, especially when applying deep learning models, has shown superior results in dynamic environments [134].

While the technical accuracy is high, the infrastructure required can involve high costs. Some systems require the installation of cameras, embedded sensors, or continuous connection to the cloud, which is not always feasible. In contrast, the use of personal devices such as smartphones or smartwatches with integrated sensors is presented as a cost-effective solution [143,145], reducing the need for specialised hardware.

The most effective systems are those capable of adapting to the conditions of the environment and the individual behaviour of the user. Models with continuous learning capability and in-line calibration enable this adaptability [87,92]. In addition, personalisation is enhanced through the use of voice assistants or the incorporation of distributed environmental sensors [128].

### 4.2. Accuracy and Reliability

Accuracy and reliability are essential pillars in fall detection systems for older adults. Although several studies have reported high levels of sensitivity and specificity in controlled settings—such as 99.05% sensitivity and 99.68% specificity using convolutional neural networks (CNNs) [113]—implementation in real-world scenarios still presents significant challenges. Factors such as improper placement of sensors, variability in human behaviour, and environmental conditions (lighting, obstacles, noise) significantly affect performance. The rate of false alarms remains a critical limitation, as everyday activities like sitting abruptly or tripping without falling can be misinterpreted as falls, leading to distrust in the system among users and caregivers [125,150].

To improve reliability, the use of hybrid approaches that integrate multiple sensors (inertial, pressure, acoustic, and visual) along with data fusion and continuous learning algorithms has been promoted [172]. These multimodal systems allow for greater adaptation to the environment and user movement patterns, reducing the incidence of false positives and improving sustained performance [87]. However, there is still a significant gap in validation with real data, as many solutions are trained with simulated drops in the laboratory. Therefore, the development of authentic, longitudinal datasets is key to achieving robust and clinically useful accuracy in everyday practice [112].

### 4.3. User Adoption and Acceptance

The adoption and acceptance of fall detection systems by older adults are determining factors for their effectiveness and sustainability over time. Although these systems have demonstrated high levels of technical accuracy, their practical success depends mainly on user experience, design ergonomics, and perceived usefulness. Numerous studies indicate that older adults are often reluctant to use wearable devices, such as belts, bracelets, or smartwatches, due to physical discomfort, cognitive difficulties, or low familiarity with technology [147,149]. In particular, the need to charge batteries frequently, manipulate complex interfaces, or remember to put on the device can create an additional burden that compromises adherence.

In addition, the perception of intrusiveness also negatively influences acceptance. For example, camera-based systems, while technically adequate, are often rejected by users who find constant video surveillance invasive, especially in private spaces such as bedrooms or bathrooms [164]. In response, recent research has proposed the use of passive, non-visual sensors (such as pressure sensors or depth cameras) that, by preserving privacy, improve the user’s willingness to keep them installed continuously [67]. On the other hand, wearable devices based on smart textiles or devices integrated into everyday clothing have shown potential to improve acceptance by offering a more natural and less disruptive experience [123].

Ease of use is also an essential component. Simple interfaces, clear feedback, and process automation (such as automatic system switching on or reconnecting) increase usability, especially for people with mild cognitive impairment. In this sense, it has been shown that user-centred design, with direct participation of older adults in the testing and feedback stages, significantly improves the positive perception of these technologies [173].

### 4.4. Privacy, Ethics and Data Security

The deployment of fall detection systems in residential and clinical contexts raises significant ethical challenges and concerns related to the privacy and security of personal data as these systems incorporate more advanced technologies, such as visual sensors, neural networks with contextual learning, and devices connected to the Internet of Things (IoT), the amount and sensitivity of the information collected increases, which requires robust data governance frameworks. The continuous collection of visual or auditory data, in particular, has been questioned due to its potential to infringe on users’ privacy, especially in private areas of the home such as bedrooms and bathrooms [164].

Systems based on computer vision, although they offer high levels of precision, are often perceived as invasive. To mitigate this problem, the use of depth cameras or infrared sensors has been proposed, which allow spatial information to be captured without capturing identifiable details of the face or environment, thus preserving an acceptable level of anonymity [67]. In addition, the use of non-visual sensors, such as pressure, sound, or vibration sensors, has been promoted as a less intrusive alternative, which can be discreetly integrated into the physical environment without compromising the dignity of the user [134].

From a digital security perspective, the storage and transmission of biometric and health data require specific protection measures. Systems should incorporate end-to-end encryption protocols, multi-factor authentication, and restricted role-based access, mainly when data is transmitted over public networks or stored in the cloud [92]. The absence of these mechanisms can make devices vulnerable to cyberattacks, which is particularly problematic in the case of health data, the exposure of which could have serious legal, social, and psychological consequences.

It is essential to consider the informed consent of users. Since many older adults may have cognitive difficulties, systems must offer clear, accessible, and tailored mechanisms to communicate what type of data is collected, how it is processed, and for what purposes it will be used [174]. Ethical decisions about the level of supervision, prolonged storage of images or physiological data, and access by third parties (such as family members or medical personnel) should be based on principles of autonomy, beneficence, and non-maleficence and counting on the favourable opinion of the corresponding ethics committee.

### 4.5. Research Gaps and Future Directions

Despite significant advances in automated fall detection using technologies such as inertial sensors, computer vision, and deep learning models, multiple research gaps persist that limit the widespread adoption and actual clinical impact of these systems. One of the most frequent limitations in the literature is the reliance on controlled environments and artificial simulations of falls, which do not faithfully reflect the movements, reactions, or contexts that characterise a real fall [112]. As a result, many systems exhibit a considerable decrease in accuracy when deployed in real-world settings, such as homes, nursing homes, or rural communities.

Another critical gap is the absence of standardised, ethically constructed public datasets that include genuine falls in natural conditions. This scarcity prevents the replicability of studies, limits comparisons between approaches, and restricts the training of robust and generalizable models. It is necessary to promote the development of collaborative repositories, built with ethical and participatory methods, that integrate multichannel signals (IMU, video, sound, pressure) and reflect the physical, functional, and cultural diversity of older adults [167].

From a technical point of view, progress is still needed in the design of models that incorporate continuous learning, that is, systems capable of autonomously adjusting to gradual changes in the user’s mobility, habits and environment. Along these lines, approaches such as online learning, transfer learning and federated learning could be especially useful to preserve the personalisation of the system without compromising privacy [87].

There is also great potential in the development of predictive systems, which cannot only detect a fall when it occurs but also anticipate the risk of falling based on gait patterns, fatigue, or physiological variations. These systems could be integrated with digital health platforms to activate preventive interventions in real time [173].

Pre-impact detection systems have emerged as a promising strategy to reduce the physical consequences of falls in older adults by anticipating the event before ground contact occurs. However, their practical implementation still presents significant challenges. Although wearable inertial sensors have demonstrated high reliability in identifying kinematic patterns that precede a fall [48,175], their performance may be affected by variability in device placement and motor heterogeneity among users. Furthermore, deep learning models—particularly hybrid architectures such as ConvLSTM—offer notable improvements over classical threshold- or SVM-based approaches by recognising subtle postural micro-transitions that are difficult to detect [176,177]. Nevertheless, their reliance on large volumes of labelled data has been partially mitigated by the availability of dedicated datasets such as KFall, which standardise the annotation of fall-onset events and enable more rigorous comparative evaluation [178]. Additionally, advances in multisensor fusion have contributed to reducing false alarms, a critical factor for end-user acceptance [179]. Overall, contemporary discourse emphasises that the effectiveness of these systems depends not solely on algorithmic performance but on the balanced integration of technical accuracy, everyday usability, validation in real-world contexts, and the preservation of autonomy and privacy for older adults.

In terms of interoperability, there is still a lack of compatibility between devices, communication protocols, and analytics platforms. Technical and regulatory standardisation will be key to facilitating scalability, integration into hospital or telemedicine infrastructures, and institutional adoption in public health systems.

From an ethical and social perspective, future research should delve into aspects such as technological justice, equity in access to these solutions, and cultural validation in diverse populations. The inclusion of older adults in the design, testing and evaluation phases not only improves the quality of the product but also ensures that the solutions effectively respond to their needs and expectations. Additionally, it is important to examine users’ experiences through qualitative studies.

### 4.6. Physiotherapy

Fall detection technologies provide physiotherapy with more accurate diagnoses and more effective clinical follow-up of older people [126,180,181]. In practice, its value lies in translating signals from inertial sensors, video/depth cameras, pressure platforms or laser sensing into clinically useful indicators on walking, balance and functional mobility, which help to stratify risk and plan interventions with an objective basis [180]. Prolonged recordings of wearable devices such as accelerometers and gyroscopes allow continuous monitoring and data-driven analysis to profile individual patterns of risk and adjust the dosage and focus of physiotherapy intervention [182,183].

Following a suspected fall event, machine learning models such as SE-DeepConvNet and other classifiers can detect and report the episode with high accuracy, enabling a timely clinical response (verification of patient status, lesion screening, temporal adjustment of therapeutic burden, and, where appropriate, coordination with caregivers) [182,184,185]. Multi-sensor and IoT integration provides additional context in a before, during and after the event, which makes it easier to understand the mechanics and circumstances of the fall and, consequently, refine rehabilitation objectives such as work on mediolateral stability, rebalancing strategies or strength and endurance training [183,185]. In parallel, wearable devices and home sensing systems support longitudinal monitoring of gait variables and physiological parameters, helpful in documenting progress, anticipating relapses, and making therapeutic adjustment decisions based on objective trends [124,186].

To integrate these systems into daily practice, it is advisable to establish clear operating rules: alert and clinical action thresholds, event verification procedures (false positive filters), criteria for readjusting intensity and type of exercise based on the patterns detected, and periodic performance review cycles (local calibration, threshold recalibration and basic accuracy audit) [182,187]. External validation under real-world conditions remains a requirement for clinical adoption [112].

Together, these technologies complement clinical assessment by offering objective risk indicators, support for post-event analysis and continuous monitoring, reinforcing decision-making and personalization of physiotherapy in the elderly population [124,126,180,182,185,186]. Likewise, the availability of multichannel repositories built with ethical criteria favours the generalisation and reproducibility of the systems [167].

## 5. Conclusions

The successful adoption of these systems depends not solely on their technical performance, but on factors such as convenience, non-intrusiveness, ease of use, and the preservation of privacy. Rejection of cameras, discomfort with portable devices, or difficulty operating technological interfaces affect adherence, especially in older adults with functional impairment. Ergonomic designs, discreet and participatory technologies, and respect for the user’s autonomy are key to promoting their acceptance.

The implementation of these systems must consider, from the outset, solid ethical frameworks, digital security policies, and informed consent mechanisms adapted to the cognitive abilities of older adults. Technologies such as non-visual sensors, end-to-end encryption, and data anonymization are essential to ensure the protection of sensitive information. In addition, it is necessary to promote standardised regulations that balance technical efficiency with the protection of user rights.

Despite technological advances, structural limitations persist that hinder the large-scale implementation of these systems. The lack of interoperability between devices, the absence of international standards, and the limited availability of real data represent obstacles to their integration in clinical and residential environments. In addition, more inclusive approaches are required that consider the cultural, functional, and socioeconomic diversity of the older adult population. The design of diagnostic solutions or devices requires multidisciplinary work in which technologists and health sciences professionals understand the same language and needs and find the right solutions. Overcoming these gaps will be essential to developing more equitable, scalable, and effective fall prevention solutions globally.

## Figures and Tables

**Figure 1 sensors-25-07423-f001:**
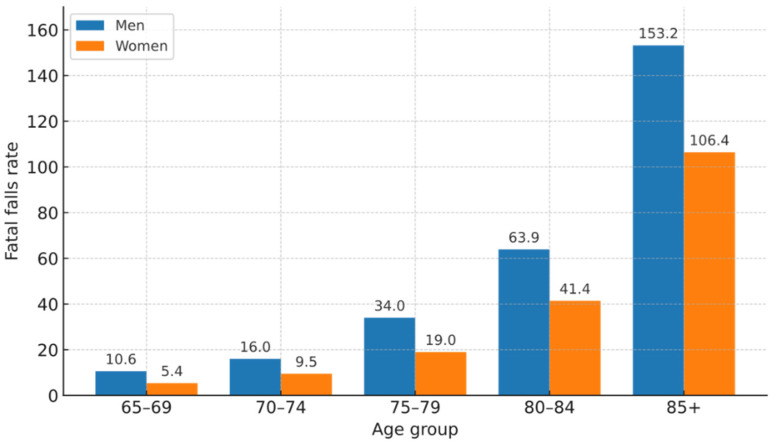
Fatal fall rate by age and sex group.

**Figure 2 sensors-25-07423-f002:**
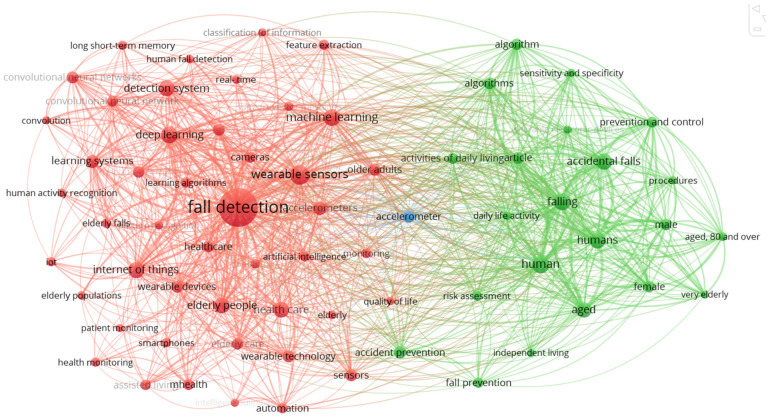
Bibliometric graph of studies on fall detection systems in older adults. Graph obtained with VOSviewer 1.6.20.

**Figure 3 sensors-25-07423-f003:**
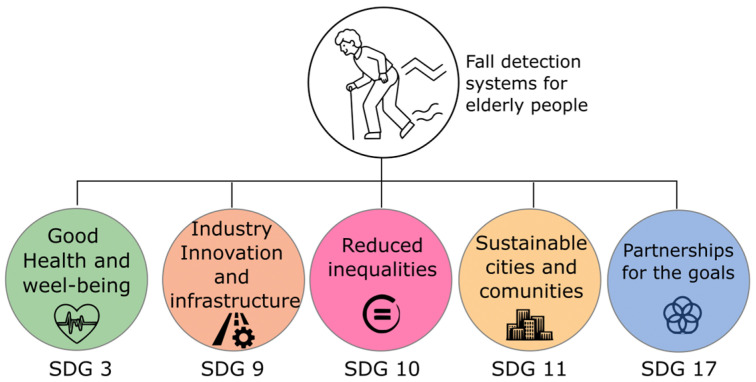
Contributions of fall detection systems for older people.

**Figure 4 sensors-25-07423-f004:**
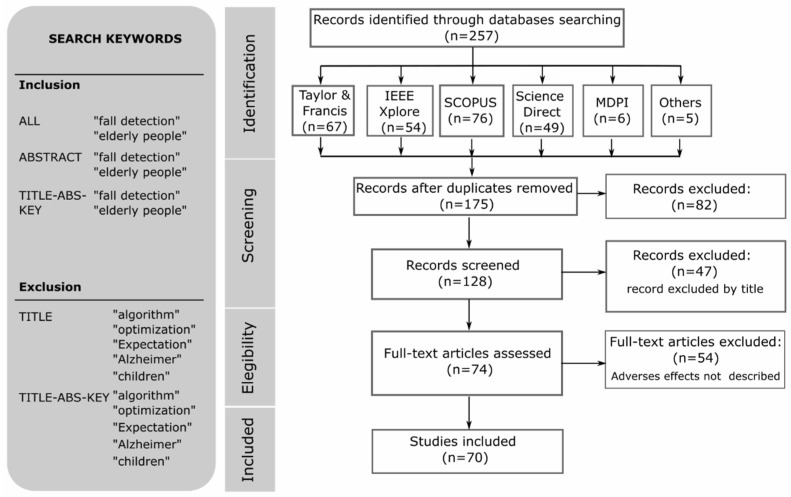
Workflow in the selection of reference documentation guideline, PRISMA^®^ methodology.

**Figure 5 sensors-25-07423-f005:**
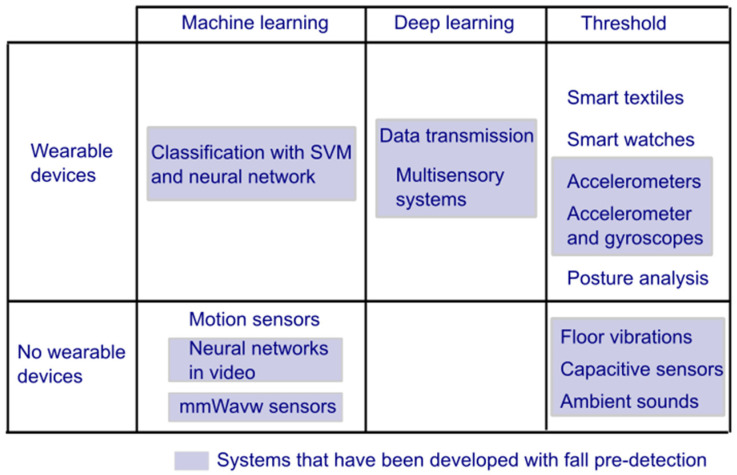
Classification of fall detection systems for older adults has been developed in the last 15 years.

**Figure 6 sensors-25-07423-f006:**
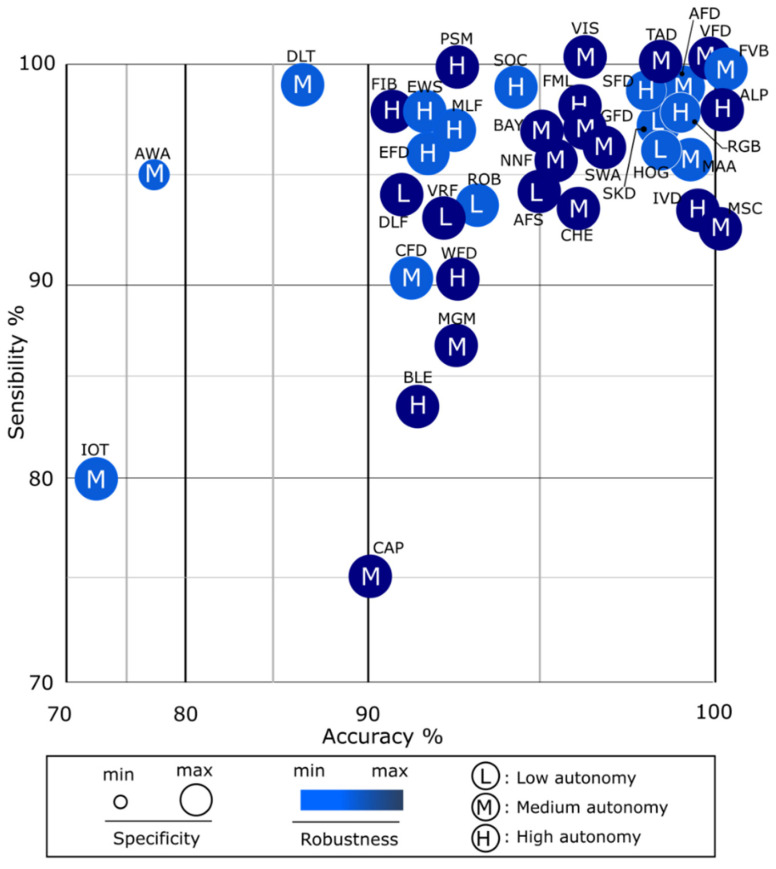
Comparison of functional parameters of fall detection systems for older adults. The abbreviations used are those shown in Table 3.

**Figure 7 sensors-25-07423-f007:**
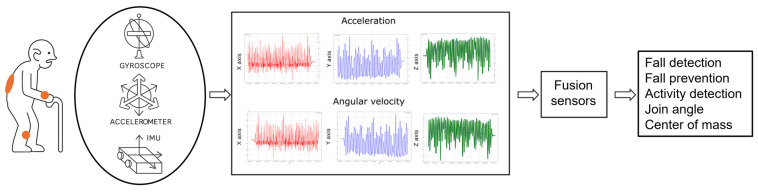
Diagram of the operation of inertial systems for fall detection.

**Figure 8 sensors-25-07423-f008:**
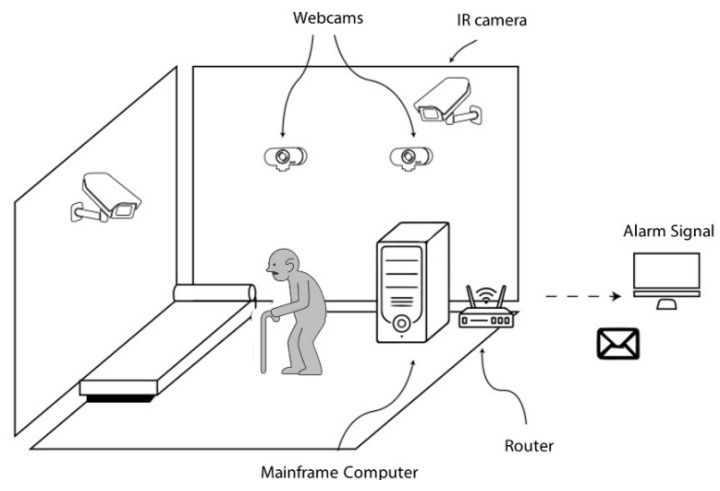
Fall detection scheme based on machine vision.

**Figure 9 sensors-25-07423-f009:**
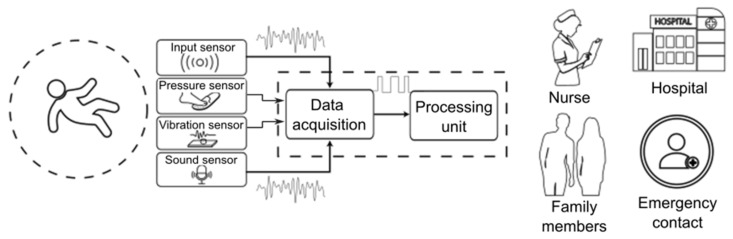
Fall detection scheme based on pressure, vibration and sound sensors.

**Figure 10 sensors-25-07423-f010:**
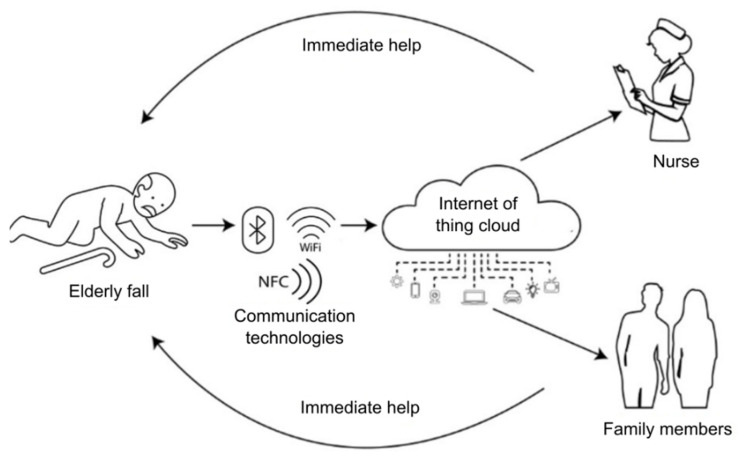
Components of fall detection systems with the use of IoT devices.

**Figure 11 sensors-25-07423-f011:**
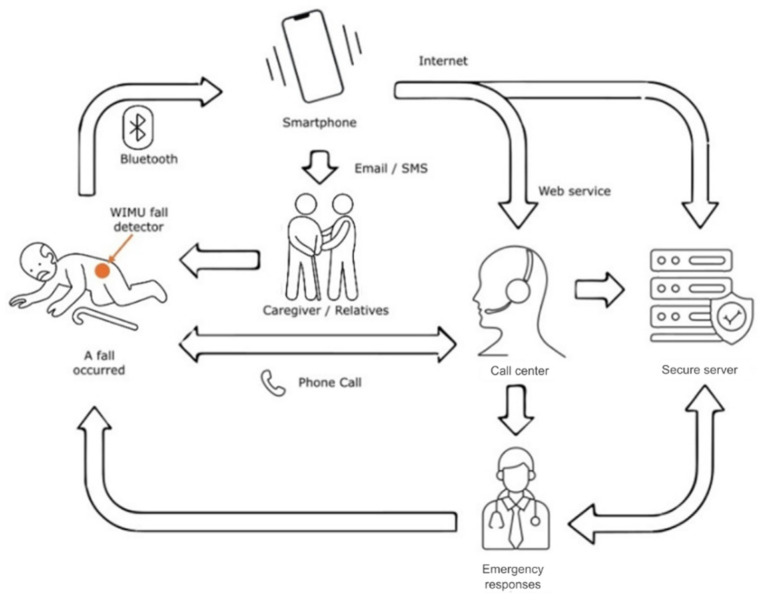
Schematic of a fall detection system based on portable devices.

**Figure 12 sensors-25-07423-f012:**
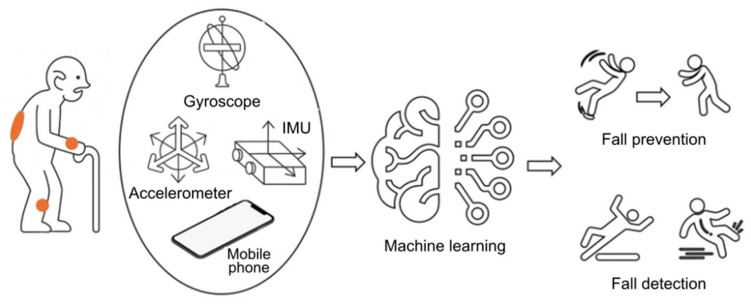
Scheme of fall detection system based on the use of machine learning and neural networks.

**Figure 13 sensors-25-07423-f013:**
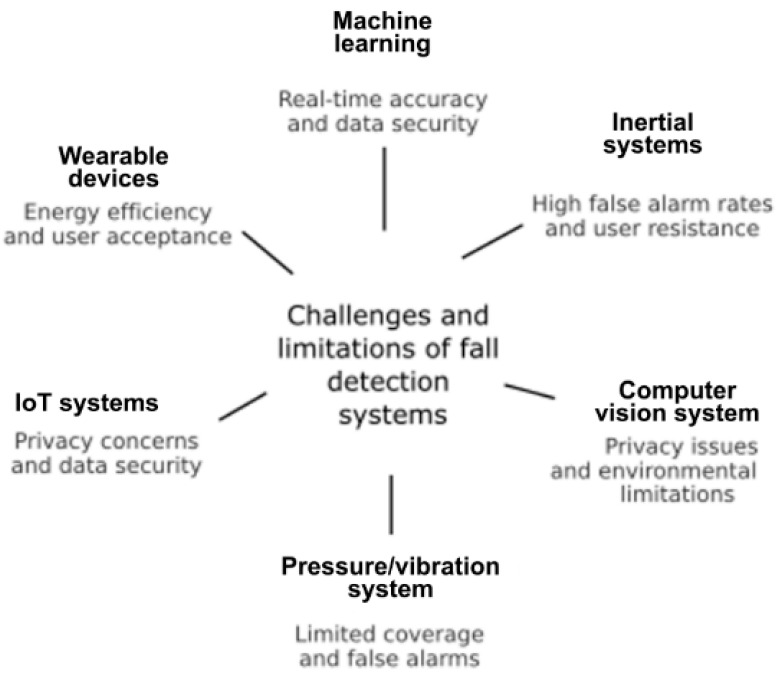
Challenges and limitations of fall detection systems in adults according to their technologies.

**Figure 14 sensors-25-07423-f014:**
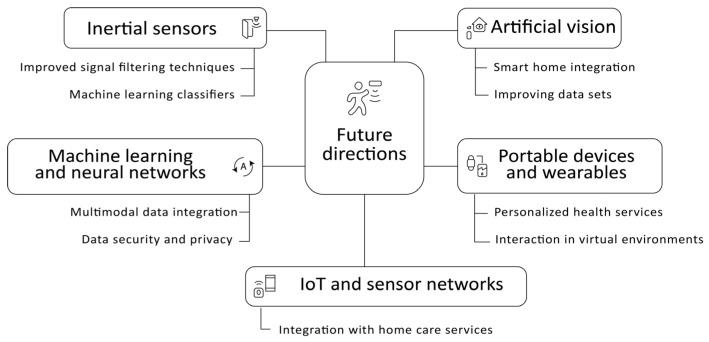
Future directions in the development of fall detection systems for elderly people.

**Table 1 sensors-25-07423-t001:** Quality assessment questions for papers.

Quality Assessment Questions	Answer
Does the document describe the technologies used in adult fall detection systems?	(+1) Yes/(+0) No
Does the document describe the operating characteristics and efficiency of fall detection systems in older adults?	(+1) Yes/(+0) No
Does the paper discuss the ethical considerations related to the use of new technologies in fall detection systems in older adults?	(+1) Yes/(+0) No
Is the journal or conference in which the article was published indexed in SJR *?	(+1) if it is ranked Q1, (+0.75) if it is ranked Q2, (+0.50) if it is ranked Q3, (+0.25) if it is ranked Q4, (+0.0) if it is not ranked

* Note: SCImago Journal Ranking (SJR).

**Table 2 sensors-25-07423-t002:** Search strings for retrieving articles in scientific literature.

Database	String	Number of Papers
SCOPUS	(TITLE (fall detection) AND TITLE (elderly people))	76
Science Direct	Title, abstract, keywords: “fall detection” AND “elderly people”	49
MDPI	“fall detection” AND “elderly people” (Topic) and Preprint Citation Index (Exclude-Database)	6
Others	Search: “fall detection” AND “elderly people” “fall detection” [All Fields] AND “elderly people” [All Fields]	5
IEEE xplore	(“Document Title”: fall detection) AND (“Document Title”: elderly people)	54
Taylor & Francis	Search: [All: “fall detection”] AND [All: “elderly people”]	67
	Total papers	257

**Table 3 sensors-25-07423-t003:** Functional parameters of fall detection systems for older adults.

Article Title	Ref	Sensitivity/Recall	Accuracy	Specificity	Robustness	Autonomy	Portability	Number of Evaluations	Number of False Positives	Real, Simulated or Controlled Environment
Inertial sensors										
Accelerometer-based fall detection using feature extraction and support vector machine algorithms (AFS).	[52]	95.00	94.58	96.70	100.00	33	100	20	3.3%	Controlled
Fall detection for older adults using machine learning (FML).	[30]	97.50	95.87	96.50	100	100	100	8	-	Controlled
A Wearable Device for Fall Detection in Elderly People Using Tri Three-Dimensional Accelerometer	[11]	100.00	-	95.68	100	100	100	-	-	Controlled
Fall detection system for older adults using IoT and Big Data (FIB).	[38]	97.5.	91.67	96.50	100	100	100	-	-	Controlled
Mobile activity recognition and fall detection system for older adults using the Ameva algorithm (MAA).	[53]	96.22	98.72	94.60	66	66	100	-	-	Semi-real
Mixing user-centred and generalised models for Fall Detection (MGM).	[46]	87.43	93.80	89.62	100	66	100	-	-	Simulated
Autonomous on-wrist acceleration-based fall detection systems: unsolved challenges (AWA).	[54]	95.45	78.12	74.62	66	66	100	-	-	Simulated
Covariance matrix-based fall detection from multiple wearable sensors (COV).	[10]	100.00	100.00	-	100	66	100	300	4%	Controlled
Chameleon: personalised and adaptive fall detection of older adults in home-based environments (CHE).	[20]	93.11	96.83	99.07	100	66	100	-	-	-
Fall event detection by gyroscopic and accelerometer sensors in a smartphone	[55]	95.00	-	96.67	66	66	100	-	-	Controlled
SVM-Based Fall Detection Method for Elderly People Using Android Low-Cost Smartphones (SFD).	[56]	99.30	98.10	96.00	66	100	100	-	-	Controlled
An EnOcean Wearable Device with a Fall Detection Algorithm Integrated with a Smart Home System (EFD).	[12]	96.00	92.10	100.00	66	100	100	100	-	Controlled
Implementation of a wireless sensor network-based human fall detection system (WFD).	[7]	98.00	90.00	93.00	100	100	100	191	12.6%	-
An Accurate Fall Detection System for Elderly People Using Smartphone Inertial Sensors (AFD).	[42]	98.88	99.27	99.66	66	66	100	-	-	Controlled
Based on computer vision (cameras and image processing)										
A video-based human fall detection system for smart homes (VFD).	[57]	100.00	100.00	93.75	100	66	33	54	6.3%	Simulated
Video Recognition of Human Fall Based on Spatiotemporal Features (VRF).	[18]	93.10	93.04	100.00	100	33	66	70	-	Controlled
An Efficient Camera-based Surveillance for Fall Detection of Elderly People (CFD).	[15]	90.00	92.50	98.93	66	66	33	-	-	Controlled
Vision-Based Fall Detection System for Improving the Safety of Elderly People (VIS).	[58]	100.00	96.66	100.00	100	66	100	-	-	Simulated
Fall detection for older adults using the variation of key points of the human skeleton (SKD).	[37]	97.00	98.50	100.00	66	33	100	-	-	Controlled
Application of k Nearest Neighbours Approach to the Fall Detection of Elderly People Using Depth-Based Sensors (KNN).	[14]	100.00	99.00	-	100	66	100	35	20%	-
A fall detection system for older adults based on integral image and histogram of oriented Gradient feature (HOG).	[9]	97.00	98.50	100.00	66	33	33	191	12.6%	Controlled
Fall detection based on the gravity vector using a wide-angle camera (GFD).	[59]	97.00	96.70	100.00	100	66	100	-	3%	Controlled
3D depth image analysis for indoor fall detection of elderly people (3DD)	[45]	100.00	98.33	-	100	66	100	-	-	Controlled
Human fall detection using machine vision techniques on RGB–D images (RGB).	[19]	97.05	98.34	97.20	66	100	100	-	-	Controlled
Fall Detection of Elderly People Using the Manifold of Positive Semidefinite Matrices (PSM).	[41]	100.00	93.38	93.00	100	100	100	-	-	Simulated
Kinect-Based Platform for Movement Monitoring and Fall-Detection of Elderly People (KIN).	[35]	100.00	98.33	-	66	33	100	-	-	Controlled
A Mobile Robot for Following, Watching and Detecting Falls for Elderly Care (ROB).	[8]	94.00	93.00	91.30	66	33	33	3996	1%	-
The implementation of an intelligent and video-based fall detection (IVD).	[16]	93.80	99.20	94.80	100	100	33	-	-	Controlled
Intelligent Elderly People Fall Detection Based on Modified Deep Learning, Deep Transfer Learning and IoT Using Thermal Imaging-Assisted Pervasive Surveillance (DLT).	[25]	99.00	86.08	99.68	66	66	100	-	-	Controlled
Based on pressure, vibration and sound sensors										
Fall detection and walking estimation using floor vibration for solitary elderly people (FVB).	[26]	100.00	100.00	98.60	66	66	33	-	-	Real
A smart capacitive measurement system for fall detection (CAP).	[60]	75.00	90.40	90.00	100	66	100	11719	0.3%	Controlled
A Framework for Fall Detection of Elderly People by Analysing Environmental Sounds through Acoustic Local Ternary Patterns (ALP).	[13]	98.00	100.00	95.00	100	100	33	10	0%	-
Based on the Internet of Things (IoT) and sensor networks										
Fall detection in older adults with mobile IoT devices and machine learning in the cloud and on the edge (IOT).	[34]	80.00	73.00	100	66	66	33	-	0	Controlled
Bluetooth-Low-Energy-Based Fall Detection and Warning System for Elderly People in Nursing Homes (BLE).	[39]	84.89	92.65	95.68	100	100	100	-	0	Controlled
Energy-efficient wearable sensor node for IoT-based fall detection systems (EWS).	[28]	98.70	92.10	81.70	66	100	100	-	-	Controlled
Towards a social and context-aware multi-sensor fall detection and risk assessment platform (MSC).	[5]	93.00	91.00	88.00	100	66	100	98	-	Controlled
Based on wearable devices										
Using a human-centred design approach to develop a fall detection sock for older women (SOC).	[31]	99.07	94.22	91.16	66	100	100	17	0	Real
Intelligent fall detection method based on accelerometer data from a wrist-worn smart watch (SWA).	[43]	96.09	97.45	98.92	100	66	100	786	9.3%	Simulated
Triaxial Accelerometer Located on the Wrist for Elderly People’s Fall Detection (TAD).	[32]	100.00	98.00	98.10	100	66	100	-	-	-
Based on machine learning models and neural networks										
Fall detection for older adults using machine learning (MLF).	[22]	98.70	92.10	81.70	66	100	100	8	-	Controlled
A Cross-dataset Deep Learning-based Classifier for People Fall Detection and Identification (DLF).	[61]	98.00	92.50	99.00	100	66	100	-	1%	Simulated
An Improved Fall Detection Approach for Elderly People Based on Feature Weight and Bayesian Classification (BAY).	[17]	95.75	95.67	98.76	100	66	100	-	1.2%	-
Accurate Fall Detection and Localisation for Elderly People Based on Neural Network and Energy-Efficient Wireless Sensor Network (NNF).	[62]	96.50	95.25	94.00	100	66	100	80	-	Controlled

**Table 4 sensors-25-07423-t004:** Features of pressure, vibration, and sound sensors in fall detection systems.

Sensor Type	Application	Advantages	Challenges
Pressure sensors	Insoles, smart carpets	High accuracy, reliable	Limited to specific environments
Vibration sensors	Floor-based systems	Effective in detecting falls and walking	Requires installation in living spaces
Sound sensors	Combined with other sensors	Enhances accuracy when combined	Not reliable alone, prone to false alarms

**Table 5 sensors-25-07423-t005:** Devices and sensors used in fall detection based on wearable devices.

Device Type	Sensors Used	Detection Method	Notification System	Accuracy	Additional Features
Smartphones, Raspberry Pi, Arduino, NodeMcu, Custom embedded systems	Accelerometer	Machine learning model	SMS, buzzer, emergency services	99.7%	Device-type invariant, real-time monitoring
Node MCU microcontroller	MPU6050 (Accelerometer, gyroscope)	Predefined thresholds	Email	Not specified	Dual-notification system
LowPAN device wearable	3D-axis accelerometer	Decision trees-based big data model	Notifications to caregivers	High success rates	Cloud services for data storage and analysis
Multiple Sensors (Wi-Fi, Floor Pressure, Smart carpets, accelerometers, gyroscopes, GPS, pulse sensors)	Various	Sensor integration	Real-time tracking	Not specified	Advanced sensing technologies
Tri-axial accelerometer, Kinect camera systems	Accelerometer, camera	SVM algorithm, PCA features	Smartphones, healthcare centres	Not specified	Cloud processing
Wearable Sensors (Gyroscope, Accelerometer)	Gyroscope, accelerometer	Data analysis	Notification system	97%	Monitoring various ADLs
Bracelet-type Device	Accelerometer	Machine learning algorithm	Mobile application	91%	Vital signs monitoring
MEMS Sensor, GSM Module, Arduino UNO	Not specified	Not specified	Nearby individuals	Not specified	Designed for wheelchair users
Wearable sensor	3-axis accelerometer	Threshold-based approach	Android application	Not specified	Cloud storage for data access
Wearable devices	Accelerometers, gyroscopes	Hybrid HSSTL optimisation model	Blockchain network	97.4%	Secure data storage, emergency response

**Table 6 sensors-25-07423-t006:** Advantages and disadvantages of fall detection systems according to the different technologies addressed in Section 3.2.

Advantages	Disadvantages
Based on inertial sensors	
High accuracy and sensitivity	False alarms and missed detections
Real-time detection and alerts	Power consumption and battery life
Portability and comfort	Data processing and computational demands
Privacy preservation	User adaptability and installation complexity
Low costs	
Based on computer vision	
Non-invasive and comfortable	Privacy issues
High precision and efficiency	High computational cost
Ease of deployment	Dependence on environmental conditions
IoT integration	Difficulties in generalisation
Robustness in different scenarios	High initial cost
Based on pressure, vibration, and sound sensors	
High accuracy	False alarms, limited scope
Non-intrusive	Installation complexity
Cost-effective	More possibility of False positives
High sensitivity	Noise interference
Non-wearable	Privacy concerns
Complementary detection	
Based on the internet of things (IoT) and sensor networks	
Enhanced safety and well-being	Usability and acceptability issues
High accuracy	Technical challenges
Non-intrusive monitoring	Privacy concerns
Integration with smart home environments	
Based on wearable devices	
Portability and ubiquity	User compliance and comfort
Enhanced detection accuracy	Battery life and maintenance
Cost-effectiveness	False alarms
Real-time monitoring and alerts	Detection limitations
	Privacy concerns
Based on machine learning models and neural networks	
High accuracy and sensitivity	Data collection
Improved fall detection ratios	Personalisation challenges
Robustness to false positives	Complexity and computational load
Data fusion and multimodal approaches	Dependence on sensor quality and placement
Low computational cost	Energy efficiency concerns
	Privacy issues

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
