# Peer review of "A Comparative Overview of Technological Advances in Fall Detection Systems for Elderly People"

_sensors, 2025, doi:10.3390/s25247423_

Round 1

Reviewer 1 Report

Comments and Suggestions for Authors

In this work, the authors present an interesting review on fall-detection technologies over the last 10–15 years. Although the importance of a comprehensive review on the theme of the manuscript is out of question, the quality of the presentation of the materials (its consistency) should be improved. I have several concerns and comments for the authors summarized below. I hope that they will help to improve the manuscript.

Comments:

  1. My main concern is the inconsistent classification for fall detection systems starting from Section 3.1. The lack of a logically clear and accurate classification makes it difficult to understand the material. Proper systematic reconsideration of the classification and a major revision of the whole material accordingly are needed. E.g., the current classification treats “Machine Learning” as a separate parallel branch but ML/DL is predominantly a processing approach used across all sensing types. It would be advisable to reorganize the classification by (a) sensor/source (inertial, vision, environmental, etc.) and/or (b) analytics layer (thresholding, classical ML, DL,etc.). This will prevent conceptual overlap and misinterpretation. Also clarify ambiguous placements (e.g., “Kinect” in Fig. 5 is a specific RGB-D/ToF sensor, not a category by itself).
  2. Table 1. Did you take into account the number of subjects for which the technologies described in the documents were tested? Some papers may describe method test results for a very limited number of subjects (~10), while others are tested on 1k+.
  3. Not all methods/technology types are reviewed in the paper. It is unclear why, for example, mm-wave sensor methods are omitted.
  4. Section 3. Choose one term and stick to it across the text, tables, and figures. If you use a classification category ‘computer vision’ or ‘wearable sensors,’ please use these terms throughout the whole document to avoid misunderstanding. Moreover, pages 10–12 mix “Portable Devices” and “Wearable Technologies,” but the distinction is not explicit. Please add brief definitions of both and then harmonize labels across the text, Figure 5, and Table 3.
  5. Table 6. It is unclear why the numbers of advantages and disadvantages are equal for one method. Most of them are shared between the methods. It may be more informative to group them in two separate tables one for advantages and another for disadvantages. In each of them, list all methods and mark adv/dis for each method. In this case it would be easy for a reader to see the whole picture across the methods.

Author Response

REVIEWER 1

We sincerely thank the reviewer for their valuable feedback and constructive suggestions, which have significantly contributed to improving the quality and clarity of our manuscript. We deeply appreciate the time and effort dedicated to providing such insightful comments. Their contributions have enriched our work and strengthened its academic rigor.

Reviewer comments

In this work, the authors present an interesting review on fall-detection technologies over the last 10–15 years. Although the importance of a comprehensive review on the theme of the manuscript is out of question, the quality of the presentation of the materials (its consistency) should be improved. I have several concerns and comments for the authors summarized below. I hope that they will help to improve the manuscript.

Comment 1

My main concern is the inconsistent classification for fall detection systems starting from Section 3.1. The lack of a logically clear and accurate classification makes it difficult to understand the material. Proper systematic reconsideration of the classification and a major revision of the whole material accordingly are needed. E.g., the current classification treats “Machine Learning” as a separate parallel branch but ML/DL is predominantly a processing approach used across all sensing types. It would be advisable to reorganize the classification by (a) sensor/source (inertial, vision, environmental, etc.) and/or (b) analytics layer (thresholding, classical ML, DL,etc.). This will prevent conceptual overlap and misinterpretation. Also clarify ambiguous placements (e.g., “Kinect” in Fig. 5 is a specific RGB-D/ToF sensor, not a category by itself).

Response 1

We have improved the classification of the systems based on your comments. This revision has indeed enhanced and clarified the categorization presented in Figure 5. Additionally, markers have been incorporated to highlight which of these systems include pre-impact fall detection, the latter in response to a recommendation from another reviewer.

Regarding the text and Table 3, the systems were classified according to the titles of the corresponding scientific articles. We believe that this approach may help readers more easily locate specific works of interest by their titles.

Focusing specifically on your observations, and with the aim of improving the structure of the text, we considered a classification based on the following categories: (1) Primary sensing technologies, (2) Perception and computer vision technologies, (3) Communication and connectivity technologies, (4) Personal-use technologies, and (5) Intelligent processing and analytical technologies. However, such a classification would also lead to cases in which, for example, inertial sensors would appear across two or more categories, generating redundancies that may also cause confusion for the reader.

Comment 2

Table 1. Did you take into account the number of subjects for which the technologies described in the documents were tested? Some papers may describe method test results for a very limited number of subjects (~10), while others are tested on 1k+.

Response 2

We have reviewed the corresponding section and have updated Table 3, which contains the characteristics of the systems, to include the number of evaluations conducted on human participants, as well as those performed using video data or simulations based on datasets specified in the respective articles. Additionally, we have incorporated the reported number of false positives and, finally, a column indicating whether the evaluations were carried out in controlled environments, real-world settings, or through simulations.

Comment 3

Not all methods/technology types are reviewed in the paper. It is unclear why, for example, mm-wave sensor methods are omitted.

Response 3

We have included, within the category of systems based on pressure, vibration, and sound sensors, those systems that employ millimeter-wave (mmWave) technology for fall detection. Their advantages have been emphasized, particularly their low intrusiveness and high spatial resolution, which make them suitable for home-based monitoring. Additionally, we highlight the enhanced performance achieved when these systems are combined with deep neural networks such as LSTM and CNN architectures optimized for real-time operation.

Comment 4

Section 3. Choose one term and stick to it across the text, tables, and figures. If you use a classification category ‘computer vision’ or ‘wearable sensors,’ please use these terms throughout the whole document to avoid misunderstanding. Moreover, pages 10–12 mix “Portable Devices” and “Wearable Technologies,” but the distinction is not explicit. Please add brief definitions of both and then harmonize labels across the text, Figure 5, and Table 3.

Response 4

This aspect has been improved. The terminology has been corrected in the text and corresponding tables to consistently use “computer vision” and “wearable devices,” ensuring that the reader does not encounter ambiguity.

Comment 5

Table 6. It is unclear why the numbers of advantages and disadvantages are equal for one method. Most of them are shared between the methods. It may be more informative to group them in two separate tables one for advantages and another for disadvantages. In each of them, list all methods and mark adv/dis for each method. In this case it would be easy for a reader to see the whole picture across the methods.

Response 5

The description of advantages and disadvantages presented in Table 6 showed an unequal number of items in each category. Initially, these were grouped to make the table appear more homogeneous; however, we have updated the table, as presenting the information in separate rows provides clearer visualization and improves reader comprehension.

Reviewer 2 Report

Comments and Suggestions for Authors

This manuscript describes an overview of technological advances in fall detection systems for the elderly. It would be helpful to understand current different applications in fall detection of the elderly people. Overall, this paper would come from the through preparation and effort to provide the technological comparisons in fall detection systems.

However, this paper, as a review paper, is felt to be somewhat lacking in certain areas and requires revisions, thus necessitating improvements and supplementation.

[Major Comments]

  1. While the paper broadly covers various research methods for fall detection, the differentiated explanation of each approach is somewhat insufficient. Beyond merely describing the general pros and cons of fall detection methods, a more specific, and if necessary, quantitative comparison is required. Furthermore, the references for each fall detection method are inadequate, leading to a perceived lack of understanding of the diverse technical approaches for each method. It is necessary to introduce a wider variety of technical approaches for fall detection systems.

  1. Since achieving 100% perfect fall detection is realistically difficult, methodological false positives (false alarms) are inevitable. A discussion on the acceptable tolerance range for these false positives is needed.

  1. Overall, the paper focuses primarily on post-impact fall detection. However, much research is currently underway on pre-impact fall detection for more proactive fall prevention. The review of the latter research in this paper is somewhat weak and needs to be supplemented. Furthermore, in the case of pre-impact falls, a lead time of only about 0.19 seconds is considered insufficient to activate protective devices such as wearable airbags to prevent serious injury. It is necessary to refer to a wider range of research findings on pre-impact fall detection.

  1. Real-time application, false alarm rates, and computing load are important factors in fall detection research, and appropriate acceptable criteria for these are needed. In particular, one of the most critical aspects of elderly fall research is the conduct of studies using real-life data, which is insufficiently addressed. Since results in real-life settings may differ from those obtained in lab environments, a specific discussion on this is deemed essential for this paper.

  1. Since several figures used in the paper are identical or similar to those in the original sources, proper citation or referencing is required.

[Minor comments]

  1. In the second paragraph of Section 2. Methodology, change "Annexe A" to "Appendix A."
  2. In Table 2, revise the column header "Nro de Papers" to "Number of Papers."
  3. In Table 3, all article titles are indicated with abbreviations like "AFS," and these abbreviations are also displayed in Figure 6, making the presentation look quite cluttered. Since a reference is provided for each paper anyway, it seems more appropriate to use the reference number instead of the abbreviation, which should also make the figure cleaner.
  4. In Figure 7, clarify the meaning of "Motion parameters" and whether grouping it with other items is appropriate. Also, correct "Join angle" to "Joint angle."
  5. The term "Input sensors" in Figure 9 is understood to encompass input sensors including pressure, vibration, and sound sensors. It is recommended that these items not be grouped together (or be clearly delineated).
  6. There are several instances of paragraphs consisting of only a single or a few sentences. It is necessary to add more sentences or integrate these paragraphs.
  7. In Section 3.3.4, correct "LT$" to "LTE."
  8. In Table 6, correct "Inercial" to "Inertial." Also, the line "Based on Inercial Sensors" appearing below "Based on Pressure, Vibration, and Sound Sensors" needs to be deleted.

Comments on the Quality of English Language

The English needs to be improved in some sentences.

Author Response

REVIEWER 2

We sincerely thank the reviewer for their valuable feedback and constructive suggestions, which have significantly contributed to improving the quality and clarity of our manuscript. We deeply appreciate the time and effort dedicated to providing such insightful comments. Their contributions have enriched our work and strengthened its academic rigor.

Reviewer comments

This manuscript describes an overview of technological advances in fall detection systems for the elderly. It would be helpful to understand current different applications in fall detection of the elderly people. Overall, this paper would come from the through preparation and effort to provide the technological comparisons in fall detection systems.

However, this paper, as a review paper, is felt to be somewhat lacking in certain areas and requires revisions, thus necessitating improvements and supplementation.

[Major Comments]

Comment 1

While the paper broadly covers various research methods for fall detection, the differentiated explanation of each approach is somewhat insufficient. Beyond merely describing the general pros and cons of fall detection methods, a more specific, and if necessary, quantitative comparison is required. Furthermore, the references for each fall detection method are inadequate, leading to a perceived lack of understanding of the diverse technical approaches for each method. It is necessary to introduce a wider variety of technical approaches for fall detection systems.

Response 1

To improve this aspect, information regarding the number of experiments conducted has been incorporated, and emphasis has been placed on whether the evaluations were performed in real-world, controlled, or simulated environments using fall simulation datasets. In addition, it has been clarified that the experiments, depending on the methodological approach, were carried out with human participants, video recordings, or test files obtained from public repositories, as specified in each of the methods discussed. Furthermore, the millimeter-wave method has been added following the recommendation of another reviewer, in order to ensure that the work reflects the most current and comprehensive information available.

Comment 2

Since achieving 100% perfect fall detection is realistically difficult, methodological false positives (false alarms) are inevitable. A discussion on the acceptable tolerance range for these false positives is needed.

Response 2

The number of false positives reported in the respective studies has been included in Table 3. In most cases, this value is not presented as an absolute count but rather as a percentage relative to the number of experiments conducted. The acceptable tolerance range for this type of application has been highlighted in the Discussion section.

Comment 3

Overall, the paper focuses primarily on post-impact fall detection. However, much research is currently underway on pre-impact fall detection for more proactive fall prevention. The review of the latter research in this paper is somewhat weak and needs to be supplemented. Furthermore, in the case of pre-impact falls, a lead time of only about 0.19 seconds is considered insufficient to activate protective devices such as wearable airbags to prevent serious injury. It is necessary to refer to a wider range of research findings on pre-impact fall detection.

Response 3

We had initially considered addressing pre-impact fall detection systems for older adults in a separate manuscript. However, in order to respond to the reviewer’s recommendation and to ensure that the present manuscript incorporates these current developments, we have highlighted in the classification presented in Figure 5 those systems that document early fall detection solutions. Additionally, at the end of Section 3.1, we have included a subsection describing the most relevant advances in pre-impact fall detection systems for older adults. Finally, the Discussion section now addresses the relevance of studying and advancing these systems, outlining their implications for future research.

Comment 4

Real-time application, false alarm rates, and computing load are important factors in fall detection research, and appropriate acceptable criteria for these are needed. In particular, one of the most critical aspects of elderly fall research is the conduct of studies using real-life data, which is insufficiently addressed. Since results in real-life settings may differ from those obtained in lab environments, a specific discussion on this is deemed essential for this paper.

Response 4

After incorporating the false alarm values into Table 3, we have included an analysis of these findings and additionally strengthened the Discussion (Section 4.2) by addressing the gap between the performance of these systems in real-world conditions. This was done in order to highlight the important point you raised.

Comment 5

Since several figures used in the paper are identical or similar to those in the original sources, proper citation or referencing is required.

Response 5

We have included, in the same sentence where the figure is first mentioned, a citation referencing a similar figure that served as the basis for redrawing the figures presented in this article.

[Minor comments]

Comment 1

In the second paragraph of Section 2. Methodology, change "Annexe A" to "Appendix A."

Response 1

The requested correction has been made.

Comment 2

In Table 2, revise the column header "Nro de Papers" to "Number of Papers."

Response 2

These terms have been corrected as suggested, which will improve the reader’s comprehension

Comment 3

In Table 3, all article titles are indicated with abbreviations like "AFS," and these abbreviations are also displayed in Figure 6, making the presentation look quite cluttered. Since a reference is provided for each paper anyway, it seems more appropriate to use the reference number instead of the abbreviation, which should also make the figure cleaner.

Response 3

We attempted to include the reference numbers in Figure 6; however, because citation numbers require the use of brackets, most of the text labels in the figure would increase from three to four characters, resulting in a more cluttered and less readable image. If this modification is mandatory, we would appreciate your guidance so that we may explore an alternative solution to address this point appropriately.

Comment 4

In Figure 7, clarify the meaning of "Motion parameters" and whether grouping it with other items is appropriate. Also, correct "Join angle" to "Joint angle."

Response 4

We replaced the term “movement parameters” with “fall detection” and additionally included “fall prevention,” which is identified in the scientific literature as one of the system outputs. The term “Join” has also been corrected to “Joint.”

Comment 5

The term "Input sensors" in Figure 9 is understood to encompass input sensors including pressure, vibration, and sound sensors. It is recommended that these items not be grouped together (or be clearly delineated).

Response 5

The figure has been improved by adding arrows to highlight the idea that these sensors transmit signals independently before they are subsequently processed, thereby enhancing the reader’s understanding.

Comment 6

There are several instances of paragraphs consisting of only a single or a few sentences. It is necessary to add more sentences or integrate these paragraphs.

Response 6

We have corrected this important aspect throughout the entire document.

Comment 7

In Section 3.3.4, correct "LT$" to "LTE."

Response 7

We have corrected this error. Thank you for your comment

Comment 8

In Table 6, correct "Inercial" to "Inertial." Also, the line "Based on Inercial Sensors" appearing below "Based on Pressure, Vibration, and Sound Sensors" needs to be deleted.

Response 8

The issue has been corrected as noted. Thank you for your comment

Round 2

Reviewer 1 Report

Comments and Suggestions for Authors

I have no further suggestions or comments as all  all points from my previous review were addressed, the content became more informative and clear. The manuscript can be accepted as it is.

Author Response

REVIEWER 1

We sincerely thank the reviewer for their valuable feedback and constructive suggestions, which have significantly contributed to improving the quality and clarity of our manuscript. We deeply appreciate the time and effort dedicated to providing such insightful comments. Their contributions have enriched our work and strengthened its academic rigor.

Reviewer comments

Comment 1

I have no further suggestions or comments as all  all points from my previous review were addressed, the content became more informative and clear. The manuscript can be accepted as it is.

Response 1

We sincerely thank the reviewer for the positive evaluation and for acknowledging the improvements made to the manuscript. We appreciate the time and effort dedicated to the review process. We are pleased to know that all previous comments were satisfactorily addressed and that the manuscript is now clear and informative.

Thank you again for your valuable contribution to strengthening the quality of our work.

Reviewer 2 Report

Comments and Suggestions for Authors

Dear Authors,

I believe that the issues I pointed out last time have been well corrected and supplemented overall. However, I think a few additional revisions below are necessary.

  1. The word "Umbralizacion" in Figure 5 looks Spanish and should be replaced with the correct English term
  2. Capitalization in titles/captions for tables (e.g., Table 3) and figures (e.g., Figure 5, Figure 6, ...) needs to be standardized.
  3. You need to provide the current title of Section 3.1.7 in Korean so I can translate it into English.
  4. The phrase "Based on Inertial Sensors" in the second line of page 20 is considered unnecessary and should be removed.
  5. The inconsistent formatting in the References section must be fixed.
Comments on the Quality of English Language

The English needs to be improved in some sentences.

Author Response

REVIEWER 2

We sincerely thank the reviewer for their valuable feedback and constructive suggestions, which have significantly contributed to improving the quality and clarity of our manuscript. We deeply appreciate the time and effort dedicated to providing such insightful comments. Their contributions have enriched our work and strengthened its academic rigor.

Reviewer comments

I believe that the issues I pointed out last time have been well corrected and supplemented overall. However, I think a few additional revisions below are necessary.

Comment 1

The word "Umbralizacion" in Figure 5 looks Spanish and should be replaced with the correct English term.

Response 1

Figure 5 has been corrected.

Comment 2

Capitalization in titles/captions for tables (e.g., Table 3) and figures (e.g., Figure 5, Figure 6, ...) needs to be standardized.

Response 2

We have verified that the comments regarding figures and tables throughout the document comply with the guidelines.

Comment 3

You need to provide the current title of Section 3.1.7 in Korean so I can translate it into English.

Response 3

The title was in Spanish, but we have now translated it into English: “Advances in early fall detection systems for older adults.”

As mentioned, the title in Korean is requested, which would be: “ 노인을 위한 초가을 감지 시스템의 발전.  “

Comment 4

The phrase "Based on Inertial Sensors" in the second line of page 20 is considered unnecessary and should be removed.

Response 4

The phrase “as mentioned” has been removed..

Comment 5

The inconsistent formatting in the References section must be fixed.

Response 5

We have verified the reference format to address this comment. We have corrected some words in Spanish and translated some that are in Chinese into English.